# A Dirac-Frenkel-Onsager Principle: Instantaneous Residual Minimization with Gauge Momentum for Nonlinear Parametrizations of PDE Solutions

Matteo Raviola [1]  Benjamin Peherstorfer [2]

## Abstract

Dirac-Frenkel instantaneous residual minimization evolves nonlinear parametrizations of PDE solutions in time, but ill-conditioning can render the parameter dynamics non-unique. We interpret this non-uniqueness as a gauge freedom: nullspace directions that leave the time derivative unchanged can be used to select better-conditioned parameter velocities. Building on Onsager's minimum-dissipation principle, we introduce a history variable—interpretable as momentum—and inject it only along the nullspace directions. The resulting Dirac-Frenkel-Onsager dynamics preserve instantaneous residual minimization, in contrast to standard regularization that can introduce bias, while promoting temporally smooth parameter evolutions. Examples demonstrate that the approach leads to increased robustness in singular and near-singular regimes.

## 1. Introduction

Neural networks and, more broadly, nonlinear parametrizations provide an expressive ansatz class for partial differential equations (PDEs), which can complement classical discretizations in, e.g., high dimensions and reduced modeling (Han et al., 2018; Raissi et al., 2019; Du & Zaki, 2021a; Zhang et al., 2025). As in classical numerical analysis, there are two broad algorithmic paradigms. Methods following the global-in-time paradigm such as PINNs (Raissi et al., 2019) train a neural network to represent the PDE solution over the entire space-time domain by minimizing a global objective. In contrast, methods following the local-in-time or sequential-in-time paradigm advance the neural-network parameters in time by computing parameter updates at each

time step from local residual minimization, which induces an evolution on the network parameters. Analogous to global and local methods in classical numerical analysis, global- and local-in-time methods for neural networks can address complementary needs; see, e.g., Zhang et al. (2025).

**Dirac-Frenkel variational principle**  In this work, we adopt the local-in-time paradigm. Many local-in-time methods build on the Dirac-Frenkel variational principle, which performs a Galerkin-type projection at each time step to select the time derivative of the parametrized solution so that the PDE residual is minimized instantaneously (Dirac, 1930; Frenkel, 1934; Lubich, 2008). The Dirac-Frenkel principle is foundational well beyond neural PDE solvers, e.g., it has a long history of being extensively used in computational chemistry (Meyer et al., 1990; Beck et al., 2000) and for dynamic low-rank approximations (Koch & Lubich, 2007; Einkemmer et al., 2025).

**Challenge: non-unique parameter dynamics**  While the Dirac-Frenkel principle yields a well-defined evolution on the level of the parametrized function, it does not necessarily induce a uniquely defined evolution of the parameters. In particular, in neural networks, distinct parameters and distinct parameter velocities may represent the same function and the same function derivative, respectively.

This ambiguity appears when the parametrization Jacobian is rank-deficient: even though the Dirac-Frenkel residual minimization condition uniquely determines the function-level time derivative, it leaves multiple admissible parameter velocities that realize it. Even when the parameter evolution is mathematically unique, fast spectral decay of the Jacobian can lead to ill-conditioning so that small perturbations due to finite numerical precision and other approximations can cause erratic parameter trajectories and a loss of robustness.

The non-uniqueness and ill-conditioning issues of the Dirac-Frenkel dynamics on the parameter level are widely recognized for neural networks (Berman & Peherstorfer, 2023; Zhang et al., 2025; Feischl et al., 2024; Chen et al., 2024) as well as for other nonlinear parametrizations (Kay, 1989; Rowan et al., 2020; Kvaal et al., 2023).

**Our contribution:  Dirac-Frenkel-Onsager principle**  We systematically address the non-uniqueness and ill-

---

[1]EPFL, Lausanne, Switzerland [2]Courant Institute of Mathematical Sciences, New York University, New York, NY, USA. Correspondence to: Matteo Raviola <matteo.raviola@epfl.ch>.

*Proceedings of the $43^{rd}$ International Conference on Machine Learning*, Seoul, South Korea. PMLR 306, 2026. Copyright 2026 by the author(s).

conditioning issue by interpreting the Dirac-Frenkel under-determination of the parameter dynamics as gauge freedom. We leverage this to select parameter velocities that promote temporal smoothness while preserving the Dirac-Frenkel optimality condition of instantaneous residual minimization.

To achieve this, we introduce an auxiliary history variable that serves as a memory of the past parameter velocities. We want the history variable to be close to a parameter velocity that reflects the present Dirac-Frenkel residual-minimizing dynamics while avoiding abrupt changes. We therefore apply Onsager's principle (Onsager, 1931) to optimally update the history variable by minimizing the mismatch with the current parameter velocity against retaining past information via a quadratic dissipation penalty on changes. The resulting history variable is a low-pass filtered version of the past minimal-norm Dirac-Frenkel parameter velocities, providing a smooth momentum-like direction. We hence interpret the history variable as a momentum variable.

Importantly, and this is a key aspect of our approach, we use the momentum variable to only fix the Dirac-Frenkel gauge. In particular, this means that the Dirac-Frenkel-determined evolution in function space that guarantees instantaneous residual minimization is unchanged and the momentum is used only along the gauge directions, i.e., the directions in the nullspace of the parametrization Jacobian. By injecting the momentum only in the nullspace directions, we select a unique parameter velocity while preserving the instantaneous residual-minimization property of Dirac-Frenkel (Galerkin optimality). This yields the proposed Dirac-Frenkel-Onsager dynamics.

**Literature review**   There is a wide range of neural PDE solver approaches building on the Dirac-Frenkel variational principle, including Neural Galerkin schemes (Bruna et al., 2024), evolutional neural networks (Du & Zaki, 2021b), and many others (Anderson & Farazmand, 2022; Berman & Peherstorfer, 2023; Finzi et al., 2023; Berman & Peherstorfer, 2024; Feischl et al., 2024). Another perspective is to view the dynamics as natural gradient descent that are applied on an energy; see, e.g., Wang et al. (2021); Hu et al. (2024); Dahmen et al. (2025) in the context of neural networks. Independent of the specific instance of how the Dirac-Frenkel variational principle is used, it suffers from the non-uniqueness and ill-conditioning issues described above. This is also noted in, e.g., Finzi et al. (2023); Feischl et al. (2024); Buchfink et al. (2024); Zhang et al. (2025).

The classical approach to cope with the non-uniqueness of the parameter dynamics is to add a regularizer, which has been systematically analyzed for the first time in (Feischl et al., 2024). However, Tikhonov regularization can introduce a bias which ought to be controlled via a residual-depending regularization parameter to avoid incurring a large error. On the other hand, the works (Haegeman et al., 2011; Koch & Lubich, 2007) fix the gauge with orthogonality conditions that are specific to matrix-factorization and tensor parameterizations, which are not applicable to generic nonlinear parametrizations such as neural networks. Another remedy that has been proposed is randomization (Berman & Peherstorfer, 2023). While this can lead to better conditioned problems (Dong et al., 2025), it is not clear what the resulting time-continuous dynamics are. Finzi et al. (2023) propose restarts, that is periodically re-training the neural network from scratch during time integration, which can lead to better minimal-norm parameter velocities; however, these can add a non-negligible cost overhead and it is unclear when exactly to perform the restarts. There is also a line of work on instantaneous residual minimization that avoids the Dirac-Frenkel variational principle altogether, for example via discretize-then-optimize and related formulations (Chen et al., 2023; Zhang et al., 2025; Chen et al., 2024; Kvaal et al., 2023). However, the resulting optimization problems to be solved at each time step can be more challenging to solve (Zhang et al., 2025).

The Dirac-Frenkel dynamics are analogous to natural gradient descent (Amari, 1998) when applied to gradient flows. In the context of neural PDE solvers, natural gradient descent with momentum is used for training PINNs in, e.g., (Müller & Zeinhofer, 2023; Goldshlager et al., 2024; Guzman-Cordero et al., 2025). Crucially, the dynamics in this case represent optimization and the previous optimization update is used as momentum rather than a moving average of the intermediate minimal-norm solutions.

**Summary of contributions**   (a) Remove a core failure mode of Dirac-Frenkel dynamics in ill-conditioned regimes while maintaining residual minimization, avoiding the bias that can be introduced by standard regularization.

(b) Address the ill-conditioning in a principled way by recasting it as gauge freedom and fixing the gauge via Onsager momentum injections along gauge directions.

(c) Improve practical reliability of neural PDE solvers by producing smoother, less erratic parameter velocities.

## 2. Dirac-Frenkel Variational Principle

### 2.1. Instantaneous residual minimization

Let $\Omega \subseteq \mathbb{R}^d$ be the spatial domain and $[0, T]$ the time interval. Consider the PDE

$$\partial_t u(t, x) = \mathcal{F}(u(t, \cdot)) \tag{1}$$

with the right-hand side operator $\mathcal{F}$ that can contain partial derivatives and initial $u_0 : \Omega \to \mathbb{R}$ and boundary conditions so that the problem is well posed over a suitable Hilbert

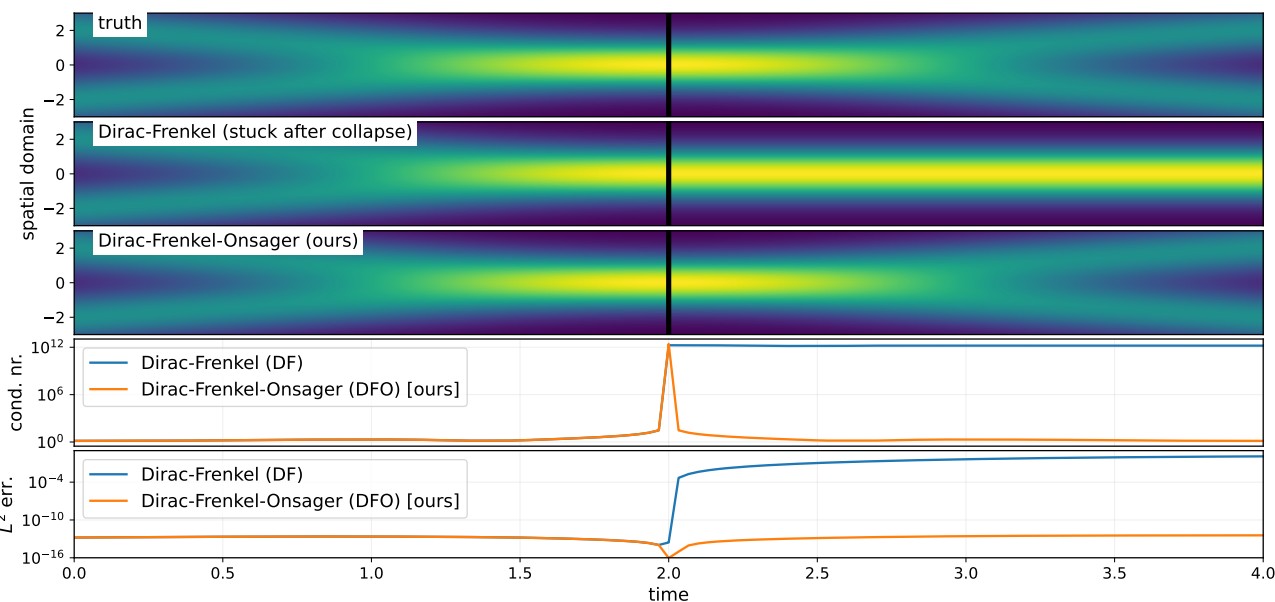

*Figure 1.* At wave collision ($t = 2$), the tangent space collapses and so Dirac-Frenkel (even with minimal-norm regularization) yields a parameter velocity that keeps the waves locked together. In contrast, the proposed Dirac-Frenkel-Onsager principle keeps injecting momentum in the nullspace direction at rank loss and so the dynamics can escape the collapse.

space $\mathcal{U}$. We aim to approximate the solution function $u : [0, T] \times \Omega \to \mathbb{R}$, which lies in $u(t, \cdot) \in \mathcal{U}$ at all times $t$.

We parametrize the solution function as $\hat{u}(\theta(t), \cdot) : \Omega \to \mathbb{R}$ with a time-dependent parameter vector $\theta(t) \in \mathbb{R}^p$ and assume that $\hat{u}$ is continuously differentiable in $\theta(t)$. In particular, the parametrization induces the trial set $\mathcal{M} = \{\hat{u}(\theta, \cdot) \,|\, \theta \in \mathbb{R}^p\}$, which is often referred to as trial manifold. The approximation $\hat{u}(\theta(t), \cdot)$ evolves in time through the time dependence of the parameter $\theta(t)$. Differentiating with respect to $t$ and applying the chain rule leads to

$$\partial_t \hat{u}(\theta(t), \cdot) = \nabla_\theta \hat{u}(\theta(t), \cdot)^\top \eta(t), \qquad (2)$$

where $\eta(t)$ is the parameter velocity and can be interpreted as the time derivative $\dot{\theta}(t)$ of the parameter vector $\theta(t)$. In particular, the time derivative (2) is an element of the space spanned by the component functions of the gradient

$$\mathcal{T}_{\hat{u}(\theta(t), \cdot)} \mathcal{M} = \mathrm{span}\{\partial_{\theta_1} \hat{u}(\theta(t), \cdot), \ldots, \partial_{\theta_p} \hat{u}(\theta(t), \cdot)\}, \qquad (3)$$

which coincides with the tangent space of $\mathcal{M}$ at $\hat{u}(\theta(t), \cdot)$ under certain regularity conditions.

Among all possible velocities at time $t$, the Dirac-Frenkel variational principle selects the residual-minimizing one

$$\partial_t \hat{u}(\theta(t), \cdot) = \underset{v \in \mathcal{T}_{\hat{u}(\theta(t), \cdot)} \mathcal{M}}{\arg\min} \|v - \mathcal{F}(\hat{u}(\theta(t), \cdot))\|^2, \quad (4)$$

where $\|\cdot\|$ denotes the $L^2$ norm over $\Omega$ in the following; see also Bachmayr et al. (2025). At the parameter level, this translates to the least-squares problem

$$\dot{\theta}(t) \in \underset{\eta \in \mathbb{R}^p}{\arg\min} \|\nabla_\theta \hat{u}(\theta, \cdot)^\top \eta - \mathcal{F}(\hat{u}(\theta(t), \cdot))\|^2. \quad (5)$$

The advantage over standard Galerkin optimality is that the Dirac-Frenkel variational principle is also applicable to parametrizations with a nonlinear parameter dependence (e.g., neural networks).

### 2.2. Conditioning and tangent space collapse

A major issue is that the induced parameter least-squares problem (5) may have a non-unique solution or be ill-conditioned, even though Dirac-Frenkel uniquely determines the residual-minimizing velocity at the function level via (4).

**Tangent space collapse** In the extreme case, the tangent map $\eta \mapsto \nabla_\theta \hat{u}(\theta, \cdot)^\top \eta$ has a non-trivial kernel. Equivalently, the tangent space has dimension strictly smaller than $p$, and we refer to this phenomenon as tangent space collapse (Zhang et al., 2025). Then the least-squares problem (5) has infinitely many minimizers: different parameter velocities $\eta$ induce the same tangent function. Choosing a particular representative, e.g., the minimum $\ell^2$-norm solution, enforces a unique parameter velocity but leads to dynamics that can be qualitatively wrong or even completely off as the following example demonstrates (see Figure 1). Even when tangent map is injective, it can be poorly conditioned (e.g., in the $L^2$ operator norm), meaning that small perturbations can produce large changes in the inferred parameter velocity. In finite precision, where time integration typically requires truncating small singular values, ill-conditioning manifests numerically as a practical tangent space collapse, which motivates methods that explicitly control how the dynamics

behave along (exact or numerical) nullspace directions.

**Example: Colliding waves** Consider the wave equation $\partial_t^2 u = c^2 \partial_x^2 u$ with $c = 1$ and initial condition $u(0, x) = \phi_0(x; -2) + \phi_\rho(x; 2)$, where $\phi_\rho(x; \mu) = \exp(-\frac{1}{2}(x - \mu)^2/(1 + \rho))$ with an offset $\rho \geq 0$. The dynamics are visualized in Figure 1 for $\rho = 0$: the two waves given by $\phi_0(x; -2)$ and $\phi_0(x; 2)$ at initial position $x = -2$ and $x = 2$, respectively, approach each other, collide at spatial location $x = 0$ and time $t = 2$, and then separate and move away from each other again.

We now demonstrate on this example analytically that applying Dirac-Frenkel with minimal-norm regularization directly can lead to dynamics which incur a large error because of the tangent space collapse. We bring the wave equation into first-order form (1) and consider the vector-valued parametrization $\hat{u}(\theta(t), x) = [\hat{u}^{(1)}(\theta(t), x), \hat{u}^{(2)}(\theta(t), x)]^\top$ with $\theta(t) = [\theta_1(t), \dots, \theta_4(t)]^\top \in \mathbb{R}^4$ and components

$$
\begin{aligned}
\hat{u}^{(1)}(\theta(t), x) &= \phi_0(x; \theta_1(t)) + \phi_\rho(x; \theta_2(t)), \\
\hat{u}^{(2)}(\theta(t), x) &= c\partial_\mu \phi_0(x; \theta_3(t)) - c\partial_\mu \phi_\rho(x; \theta_4(t)).
\end{aligned} \tag{6}
$$

As long as $\theta_1(t) \neq \theta_2(t)$, the space (3) spanned by the component functions of the gradient is four dimensional and Dirac-Frenkel leads to a unique parameter velocity $\dot{\theta}(t)$. However, when the waves collide $t = 2$, we have $\theta_1(t) = \theta_2(t)$ and the tangent space collapses to dimension two. In particular, to separate the two waves at $t > 2$, the parameter velocity needs to be aligned with the direction $[1, -1, 0, 0]^\top$, which is in the orthogonal complement of the collapsed tangent space. Thus, following the minimal-norm Dirac-Frenkel dynamics can never lead to a separation of the waves and the waves are stuck for all $t > 2$, as shown in Figure 1. Analogously, regularization with Tikhonov and truncated SVD also prevents motions in the orthogonal complement of the collapsed space. Consequentially, the two waves stick together rather than cross, leading to the collapse of even the regularized Dirac-Frenkel dynamics. See Section A.1 for additional details on this problem.

# 3. Leveraging the Gauge Freedom

## 3.1. Gauge freedom of Dirac-Frenkel dynamics

We propose to treat the non-uniqueness of the parameter velocity as a gauge freedom of the Dirac-Frenkel dynamics. To make this gauge freedom precise, it is convenient to work with the normal equations

$$
G(\theta(t))\eta(t) = g(\theta(t)) \tag{7}
$$

of the least-squares problem (5), with the matrix $G(\theta(t)) \in \mathbb{R}^{p \times p}$ and the right-hand side vector $g(\theta(t)) \in \mathbb{R}^p$ defined

as $(i, j = 1, \dots, p)$

$$
\begin{aligned}
G_{ij}(\theta(t)) &= \langle \partial_{\theta_i} \hat{u}(\theta(t), \cdot), \partial_{\theta_j} \hat{u}(\theta(t), \cdot) \rangle, \\
g_i(\theta(t)) &= \langle \partial_{\theta_i} \hat{u}(\theta(t), \cdot), \mathcal{F}(\hat{u}(\theta(t), \cdot)) \rangle,
\end{aligned}
$$

where $\langle \cdot, \cdot \rangle$ denotes the $L^2$ inner product on $\Omega$. The set of all solutions of the normal equations (7), and thus of the least-squares problem (5), is an affine space. Let $\bar{\eta}(\theta(t)) \in \mathbb{R}^p$ be a reference solution of (5), which will be the minimal-norm solution in the following, but any other reference solution can be used. All parameter velocities $\eta(t)$ that are compatible with the Dirac-Frenkel dynamics, in the sense that they solve (5), can be represented as

$$
\eta(t) = \bar{\eta}(\theta(t)) + w, \qquad w \in \text{null}(G(\theta(t))), \tag{8}
$$

where $\text{null}(G(\theta))$ denotes the nullspace of the matrix $G(\theta(t))$. Notice that the latter is the same space as the kernel of the linear map $\eta \mapsto \nabla_\theta \hat{u}(\theta(t), \cdot)^\top \eta$ induced by the gradient of $\hat{u}$ at $\theta(t)$.

We refer to this freedom of adding vectors $w$ from the nullspace $\text{null}(G(\theta(t)))$ to any parameter velocity $\bar{\eta}(\theta(t))$ as gauge freedom of the Dirac-Frenkel dynamics. In particular, the nullspace captures parameter velocity directions that are not violating the instantaneous residual minimization principle given by Dirac-Frenkel and thus are not changing the time derivative $\partial_t \hat{u}$ given in (2).

## 3.2. Gauge fixing for Dirac-Frenkel dynamics

We propose to fix the gauge of Dirac-Frenkel dynamics. This means we maintain the instantaneous residual minimization property of Dirac-Frenkel dynamics while selecting one specific parameter velocity $\dot{\theta}(t)$ in the set of all possible solutions given by (8).

One approach to fixing the gauge freedom is via an objective $\Phi : \mathbb{R}^p \times \mathbb{R}^p \to \mathbb{R}$ and an optimization problem

$$
\dot{\theta}(t) = \bar{\eta}(\theta(t)) + \underset{w \in \text{null}(G(\theta(t)))}{\arg\min} \Phi(\bar{\eta}(\theta(t)), w), \tag{9}
$$

where $\Phi$ selects one $w$ out of all possible choices in the nullspace $\text{null}(G(\theta(t)))$ (recall that $\bar{\eta}(\theta(t))$ is the reference velocity). We stress that (9) applies changes to the residual-minimizing velocity $\bar{\eta}(\theta(t))$ only in the directions of the nullspace, so that $\dot{\theta}(t)$ is also a solution of the Dirac-Frenkel least-squares problem (5) and thus an instantaneous residual minimizing velocity. This is in contrast to standard regularization that applies the regularizer to all components (within and outside of the nullspace) so that the resulting velocity is not necessarily residual-minimizing. Instead, we advocate for maintaining the residual-minimization property and using the gauge freedom to pick one parameter velocity that has favorable properties. The next section discusses one specific objective, but we remark that equation (9) should be

understood as a flexible gauge-fixing template rather than the specific Onsager-history objective function introduced below. Other choices of $\Phi$, including potentially history-free gauge-fixing rules, are possible and are an interesting direction for future work.

**First versus higher-order changes**  We note that the Jacobian nullspace characterizes directions that do not change the represented function to first order; for nonlinear parametrizations, motion in these directions can still alter the function through higher-order effects. Consequently, for finite step sizes, nullspace injection can introduce higher-order drift that is not controlled by the instantaneous Dirac-Frenkel optimality condition—indeed, this is in agreement with the overall Dirac-Frenkel principle that constrains only the first-order (tangent-space) residual minimization and imposes no variational control over higher-order effects induced by parameter motion.

# 4. Dirac-Frenkel-Onsager Dynamics

## 4.1. Onsager's principle

We introduce a history variable $m(t) \in \mathbb{R}^p$, which can be interpreted as a proposed velocity based on previous parameter velocities. Define the tracking energy

$$\mathcal{E}(m, t) = \frac{1}{2}\big\| m - \bar{\eta}(\theta(t)) \big\|_2^2,$$

which penalizes mismatch between $m(t)$ and the reference $\bar{\eta}(\theta(t))$, and define a quadratic dissipation potential $\Psi(v) = \frac{\tau}{2}\|v\|_2^2, \tau > 0$. We then apply Onsager's minimum-dissipation principle (Onsager, 1931) to $m(t)$ as

$$\dot{m}(t) \in \arg\min_{\phi \in \mathbb{R}^p} \Big\{ \Psi(\phi) + \langle \nabla_m \mathcal{E}(m(t), t),\, \phi \rangle \Big\}, \quad (10)$$

so that the stationary condition of (10) yields the linear filter

$$\tau\,\dot{m}(t) = -\nabla_m \mathcal{E}(m(t), t) = \bar{\eta}(\theta(t)) - m(t). \quad (11)$$

The explicit representation of the filter is an exponentially weighted history of $\bar{\eta}(\theta(t))$,

$$m(t) = \mathrm{e}^{-\frac{t-t_0}{\tau}}\, m(t_0) + \frac{1}{\tau}\int_{t_0}^t e^{-\frac{t-s}{\tau}}\, \bar{\eta}(\theta(s))\mathrm{d}s,$$

which reveals how the whole history of the reference $\bar{\eta}$ is included in $m$. In particular, when $\bar{\eta}(\theta(t)) = \bar{\eta}$ is fixed, then the mismatch energy decays monotonically since $\frac{d}{dt}\mathcal{E}(m(t), t) = -\frac{1}{\tau}\|m(t) - \bar{\eta}\|_2^2 \leq 0$, which demonstrates the inertia-like effect of the variable $m(t)$ and that $m(t)$ is a relaxational memory mechanism towards $\bar{\eta}$.

## 4.2. Gauge fixing with Onsager dynamics

**Momentum in orthogonal complement**  Given the reference parameter velocity $\bar{\eta}(\theta(t))$ (in our case the minimal-norm solution to (5)), we require the parameter velocity $\dot{\theta}(t)$ to match $m(t)$ imposing

$$\dot{\theta}(t) = \bar{\eta}(\theta(t)) + \arg\min_{w \in \mathrm{null}(G(\theta(t)))} -\langle m(t), w \rangle + \frac{1}{2\lambda}\|w\|_2^2, \quad (12)$$

where $m(t)$ is given by the Onsager equation (11) and $\lambda > 0$ is a regularization parameter that controls how much momentum is added along the nullspace directions.

We can write down the dynamics of $\theta(t)$ and $m(t)$ directly. Let $P(\theta) \in \mathbb{R}^{p \times p}$ be the projector onto the nullspace $\mathrm{null}(G(\theta))$, then the solution $\dot{\theta}(t)$ of (12) is

$$\dot{\theta}(t) = \bar{\eta}(\theta(t)) + \lambda P(\theta(t))m(t). \quad (13)$$

Combining (13) with the Onsager dynamics given in (11), we obtain

$$\dot{m}(t) = \frac{1}{\tau}(\bar{\eta}(\theta(t)) - m(t)), \quad (14)$$

$$\dot{\theta}(t) = \bar{\eta}(\theta(t)) + \lambda P(\theta(t))m(t). \quad (15)$$

**Discussion**  In the situation that the Gram matrix $G(\theta(t))$ is of low rank ("tangent space collapse"), the nullspace $\mathrm{null}(G(\theta(t)))$ enlarges, and so the projection term $P(\theta(t))m(t)$ becomes the mechanism that injects motion along newly available nullspace (gauge) directions. Additionally, in case of near collapse (i.e., poor conditioning of $G(\theta)$), truncating small singular values and injecting the momentum $m(t)$ in the resulting approximated nullspace acts as a low-pass filter that can prevent jumps in these directions that can be caused by numerical artifacts due to poor conditioning.

Let us also comment on how the use of momentum in our setting fundamentally differs from optimization approaches. For an optimization task, only the final/best iterate is of interest. In our case, we approximate a dynamical system which must be matched at all times $t$. This means that a naive application of momentum fails as it does not respect the instantaneous-minimization principle.

## 4.3. Time discretization

Let us now discretize the Dirac-Frenkel-Onsager dynamics with an Euler method for ease of exposition; other time-integration schemes can be applied as in Appendix E.

Consider the time domain $[0, T]$ with $0 = t_0 < t_1 < \cdots < t_K = T$ with equidistant time-step size $\delta t > 0$. We denote the reference velocity at time $t_k$ as $\bar{\eta}_k = \bar{\eta}(\theta_k) \in \mathbb{R}^p$ and the corresponding momentum variable as $m_k \in \mathbb{R}^p$.

For the Onsager evolution (11), we use backward (implicit) time stepping because it preserves the qualitative relaxation property of the Onsager filter for any time-step size $\delta t > 0$, whereas explicit methods would lead to a step-size restriction. For the parameter evolution (15) we instead employ forward (explicit) Euler, so that the combined scheme is semi-implicit. With these choices, we can exploit the Onsager evolution special structure in $m(t)$ to obtain backward Euler steps which require no solve and thus implicit time integration can be applied efficiently. More precisely, we have

$$\tau \frac{m_{k+1} - m_k}{\delta t} = \bar{\eta}_k - m_{k+1} \, ,$$

which is equivalent to the exponential moving average

$$m_{k+1} = \beta \, m_k + (1 - \beta) \, \bar{\eta}_k, \qquad \beta = \frac{\tau}{\tau + \delta t}.$$

Note that $\beta \in (0,1)$ for any $\delta t > 0$, so $m_{k+1}$ is always a convex combination of $m_k$ and $\bar{\eta}_k$. For the parameter equation, the time discretization yields

$$\frac{\theta_{k+1} - \theta_k}{\delta t} = \bar{\eta}_k + \lambda P(\theta_k) m_{k+1},$$

with the minimal-norm solution $\bar{\eta}_k = G(\theta_k)^+ g(\theta_k)$. Notice that backward in time would lead to a nonlinear problem at each time step.

Putting this together, we obtain the time-discrete dynamics with the minimal-norm parameter velocity as reference

$$\begin{aligned} \bar{\eta}_k &= G(\theta_k)^+ g(\theta_k), \\ m_{k+1} &= \beta m_k + (1 - \beta)\bar{\eta}_k, \\ \theta_{k+1} &= \theta_k + \delta t(\bar{\eta}_k + \lambda P(\theta_k) m_{k+1}). \end{aligned} \qquad (16)$$

### 4.4. Collocation and sampling in space

We introduce a set of collocation points $\{x_i\}_{i=1}^N \subset \Omega$. The collocation points can be uniformly sub-sampled from a grid in the spatial domain $\Omega$ in low dimensions, or sampled in some other way for high-dimensional problems (Wen et al., 2024). In particular, the collocation points can change over the time step, which we disregard for notational ease.

We define the batch gradient $J(\theta) \in \mathbb{R}^{N \times p}$ of $\hat{u}(\theta, \cdot)$, which we hereafter refer to as Jacobian, and the batch right-hand side $f(\theta) \in \mathbb{R}^N$ as

$$J_{ij}(\theta) = \partial_{\theta_j} \hat{u}(\theta, x_i), \qquad f_i(\theta) = \mathcal{F}(\hat{u}(\theta, \cdot))(x_i) \quad (17)$$

for $i = 1, \ldots, N$ and $j = 1, \ldots, p$. An approximation of the minimal-norm solution of the Dirac-Frenkel least-squares problem (5) can then be computed via an SVD of the batch gradient $J(\theta)$: Let $J(\theta) = U\Sigma V^\top$ be the SVD of $J(\theta)$ and let $U_\epsilon \Sigma_\epsilon V_\epsilon^\top$ be the one with all singular values below a tolerance $\epsilon > 0$ truncated (tSVD). The minimal-norm solution up to the tolerance $\epsilon$ is then formally given

---

**Algorithm 1** Dirac-Frenkel-Onsager dynamics
___
Fit parameterization $\hat{u}(\theta, \cdot)$ to initial condition $u_0$ to obtain $\theta_0$ and set $m_0 = 0$
**for** $k = 0$ **to** $K - 1$ **do**
  Assemble $J_k = J(\theta_k)$ and $f_k = f(\theta_k)$ defined in (17)
  Compute $U_\epsilon, \Sigma_\epsilon, V_\epsilon^\top = \text{tSVD}(J_k, \epsilon)$
  Compute minimal-norm solution $\bar{\eta}_k$ of (5) to tol. $\epsilon$
  Update momentum $m_{k+1} = \beta m_k + (1 - \beta)\bar{\eta}_k$
  Project momentum $m_{k+1}^\perp = m_{k+1} - V_\epsilon(V_\epsilon^\top m_{k+1})$
  Update parameters $\theta_{k+1} = \theta_k + \delta t \left( \bar{\eta}_k + \lambda m_{k+1}^\perp \right)$
**end for**
___

by $\bar{\eta} = V_\epsilon \Sigma_\epsilon^{-1} U_\epsilon^\top f(\theta)$. Furthermore, the action of the projector $P_\epsilon(\theta)$ can be computed from the same SVD as $P_\epsilon(\theta)z = z - V_\epsilon V_\epsilon^\top z$. We note that a randomized SVD can be used; see Appendix D.

### 4.5. Algorithm and costs

In Algorithm 1 we describe the DFO steps with the semi-implicit Euler scheme introduced in Section 4.3. Notice how injecting momentum requires only two extra matrix-vector multiplies compared to DF with an SVD-based least-squares solver, which amounts to a few extra lines of code. The dominant cost is the computation of the tSVD which scales like $O(Np^2)$. If we use randomized tSVD with sketch size $s$, the cost can be reduced to $O(Ns^2)$ (see Appendix D).

## 5. Numerical Experiments

We demonstrate our method on various low and high dimensional examples. In this section, we use DF to denote the Dirac-Frenkel dynamics with minimal-norm solution of the normal equations (7) obtained with truncated SVD to some tolerance $\epsilon$, unless stated otherwise.

### 5.1. Examples with tangent-space collapse

We revisit the wave example introduced earlier. Using the minimal-norm velocity as reference and the DFO dynamics given by (14)–(15) we obtain with the momentum $m(t)$ a non-zero velocity component in the direction $[1, -1, 0, 0]^\top$, which is in the Jacobian nullspace when the waves collide, and so the waves can separate after the tangent space collapse; see Figure 1. This confirms the theoretical analysis for the exact collapse case $\rho = 0$ in Section A.1 which proves that the continuous-time DFO dynamics recover the exact PDE solution through the collision for a suitable choice of $\lambda$. Additionally, we consider an ill-conditioned case with $\rho$ small but positive. In this case, DF recovers after a while but an order of magnitude larger error is attained than with DFO; see Figure 2a.

Let us consider another example affected by tangent space

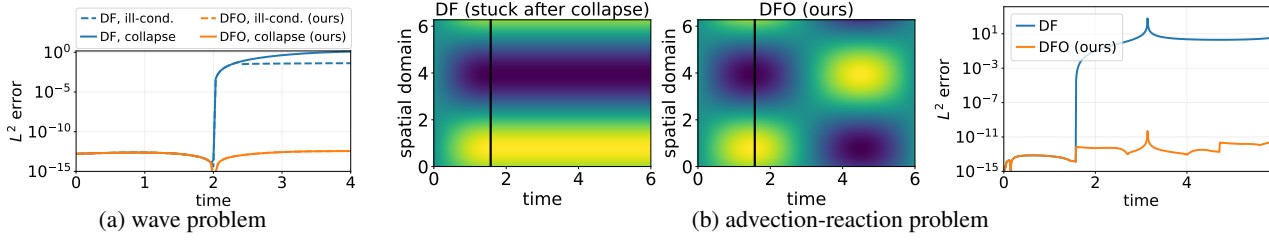

*Figure 2.* Plot (a) shows that DFO achieves orders of magnitude lower errors than DF in a severely ill-conditioned wave problem without exact collapse. Plot (b) shows that for an advection-reaction problem with repeated tangent-space collapse points at $t \in \{\pi/2, 3\pi/2\}$, the DFO momentum injects nullspace parameter velocities so that the full-dimensional tangent space is restored after collapse and parameter dynamics can continue to evolve, while DF remains stuck after the first collapse point.

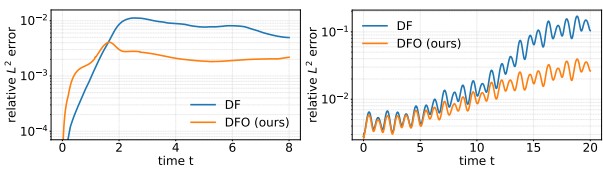

(a) rotating detonation waves  (b) transport through flow field

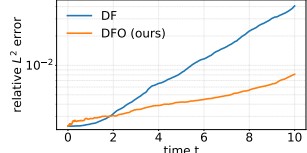

(c) charged particles in electric field

*Figure 3.* Relative error over time for the three low-/moderate-dimensional PDEs, comparing minimal-norm DF (tSVD) and the proposed DFO.

collapse: an advection-reaction equation that leads to Jacobian rank loss at time $t = \pi/2$; see Figure 2b and Appendix A.2 for details on the problem. We obtain a similar effect with the DFO dynamics (14)–(15) to the one observed for the wave equation: at the singular point, where the dimension of the tangent space drops to zero, the momentum injects non-zero velocity directions without sacrificing the instantaneously residual minimizing condition. The variational space hence recovers full dimension again, and the dynamics are approximated with high accuracy.

### 5.2. Low-dimensional PDE examples

**Examples**    (Details in Appendix B).

(a) *Rotating detonation waves*: We use the model of Koch et al. (2020), motivated by rotating detonation engines for space propulsion. Its degrees of freedom are the intensive property of the working fluid and the combustion progress. The solution features a sharp-fronted wave that propagates around a circular domain and that spawns a second wave.

(b) *Transport through flow field*: We model transport through a nonuniform, time-dependent flow field with the advection equation.

(c) *Charged particles in electric field*: We use the Vlasov equation to model the dynamics of charged particles in a collisionless plasma. Starting from a localized particle distribution, the electric field advects, shears, and deforms the density in phase space over time.

**Parametrizations, time discretization, other methods**
We use multi-layer perceptron (MLP) parametrizations with periodic embeddings to enforce periodic boundary conditions. The DF dynamics are discretized with Runge-Kutta 4 (RK4) and the DFO dynamics with a combination of RK4 and backward Euler (see Appendix E for details). We compared our approach DFO to several other methods that implement instantaneous residual minimization: Dirac-Frenkel (DF) with tSVD-based approximate minimal-norm solution (DF tsvd) as used in, e.g., Bruna et al. (2024); DF with Tikhonov regularization (DF Tiko) as in Feischl et al. (2024), TENG (Chen et al., 2024), which performs an inner re-training loop with natural gradient descent; Neural IVP (NIVP) (Finzi et al., 2023), which re-trains the neural-network from scratch at occasional time steps to improve conditioning; and RSNG (Berman & Peherstorfer, 2023), which uses randomized updates to improve conditioning. We sweep the hyper-parameters and show the best results, except for RSNG for which we show the average over five runs as it is a randomized method.

**Results: Lower overall error, negligible overhead**    Figure 3 shows that DFO leads to lower errors than minimal-norm Dirac-Frenkel dynamics. A comparison to other methods provides further evidence that Dirac-Frenkel-Onsager leads to competitive errors that are lowest across all three examples (Table 1). In particular, we note that the extra costs incurred by Onsager dynamics and the momentum variable are almost negligible, compared to the extra costs incurred by, e.g., TENG in certain problems. We note that DFO can be combined with randomization techniques to further reduce costs; see Appendix D and the high-dimensional problem below.

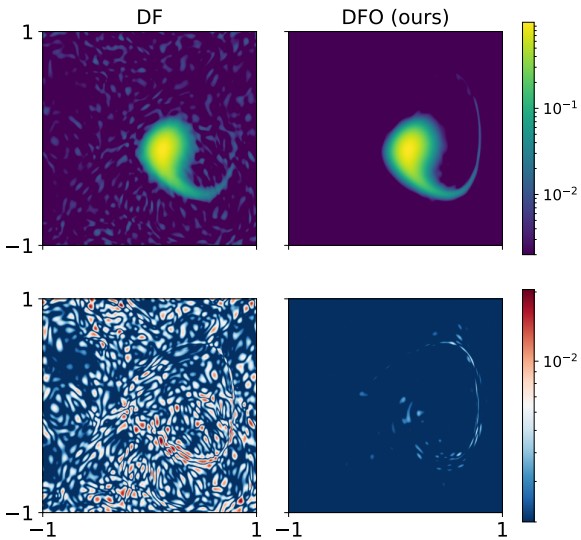

*Figure 4.* Charged particles: The pointwise error plots show that DFO (right) suppresses background error away from the solution support by using momentum to select coherent, smooth parameter velocities in nullspace directions, whereas DF (left) uses the dynamics-agnostic 2-norm regularizer that leads to less informative dynamics into the nullspace directions.

*Table 1.* DFO achieves the lowest avg. $L^2$ error with near-negligible additional cost.

| Method | $L^2$ err. | $L^2$ err. @ $T$ | Runtime (DF=1) |
|---|---|---|---|
| **Rotating detonation waves** | | | |
| DF + Tiko | 6.41e-03 | 5.45e-03 | 1× |
| DF + tSVD | 6.06e-03 | 4.02e-03 | 1× |
| NIVP | 6.41e-03 | 5.71e-03 | 1× |
| RSNG | 1.22e-02 | 4.04e-03 | **0.3×** |
| TENG | 6.83e-03 | 6.35e-03 | 3× |
| DFO (ours) | **2.10e-03** | **1.91e-03** | 1× |
| **Transport through flow field** | | | |
| DF + Tiko | 6.44e-02 | 1.07e-01 | 1× |
| DF + tSVD | 3.82e-02 | 1.05e-01 | 1× |
| NIVP | 1.69e-02 | **2.32e-02** | 1× |
| RSNG | 3.13e-02 | 1.03e-01 | 0.5× |
| TENG | 5.68e-02 | 1.55e-01 | 2× |
| DFO (ours) | **1.23e-02** | 2.63e-02 | 1× |
| **Charged particles in electric field** | | | |
| DF + Tiko | 8.25e-03 | 1.62e-02 | 0.9× |
| DF + tSVD | 1.22e-02 | 4.06e-02 | 1× |
| NIVP | 8.48e-03 | 1.79e-02 | 1× |
| RSNG | 2.72e-02 | 1.02e-01 | 0.5× |
| TENG | 6.01e-02 | 3.17e-01 | 2× |
| DFO (ours) | **3.95e-03** | **8.06e-03** | 1× |

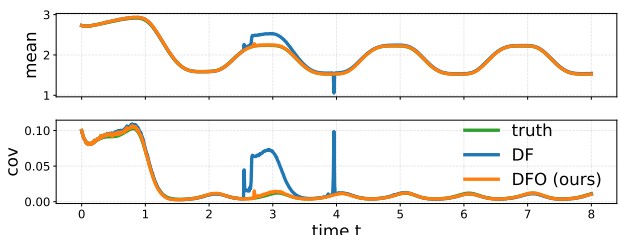

*Figure 5.* Fokker-Planck 5D: DFO, by promoting temporal smoothness and so reducing erratic parameter velocities, yields more accurate mean and covariance predictions and avoids the jumps observed in DF.

*Table 2.* Summary of results for the 5D Fokker-Planck equation.

| Method | Err. mean | Err. cov | Runtime (DF=1) |
|---|---|---|---|
| DF + tSVD | 1.46e-02 | 3.17e-01 | 1 |
| RSNG | 2.81e-02 | 4.33e-01 | **0.4×** |
| NIVP | 2.82e-01 | 1.71e+01 | 2× |
| TENG | 2.23e-02 | 5.81e-01 | 2× |
| DFO (ours) | **4.47e-03** | **5.26e-02** | 1× |
| RDFO (ours) | 4.51e-03 | 5.60e-02 | 0.5× |

## 5.3. High-dimensional Fokker-Planck equation

We consider the evolution of five interacting particles in an aharmonic confining potential with an additional repulsive interaction term whose joint density $u(t, x)$ with $x \in \mathbb{R}^5$ evolves according to a Fokker–Planck equation. We solve the Fokker–Planck equation using an MLP with three layers and width 20; see Appendix B.4.

Figure 5 shows the predicted mean and diagonal entry of the covariance corresponding to dimension five. While DF dynamics lead to jumps in the solution, our DFO leads to smoother mean and covariance predictions. This is in agreement with the results shown in Table 2, where our approach achieves the smallest approximation error for the mean and covariance. We additionally show in Table 2 a variant RDFO of DFO that uses a randomized SVD (see Appendix D for details), which helps to reduce the runtime in this example by a factor two compared to DFO and DF.

**Results: Less background error away from support** When plotting the point-wise error over the spatial domain (see Figure 4), we observe that DFO produces noticeably lower background error in regions where the true solution is essentially inactive. This improvement is explained by how DFO resolves the DF gauge freedom. The momentum supplies a coherent, temporally smooth choice for the parameter velocity components in all directions, rather than leaving the nullspace components to be set by the dynamics-agnostic 2-norm regularizer. As a result, spurious parameter motion that does not affect the solution is nonetheless steered in a consistent way, reducing the accumulation of small off-support artifacts.

# 6. Conclusions and Limitations

*Conclusions* This work proposes an Onsager-type momentum mechanism that systematically handles tangent-space collapse and ill-conditioning in Dirac-Frenkel dynamics. By injecting momentum only along Jacobian-null (gauge) directions, we uniquely determine a parameter evolution while leaving the Galerkin-optimal, instantaneous residual-minimizing function-space dynamics unchanged. Numerical experiments demonstrate that the proposed Dirac-Frenkel-Onsager scheme can cope with tangent-space collapse, reduces background error by selecting informed parameter velocities in all directions, incurs little additional cost, and scales to high-dimensional problems.

*Limitations* (i) DFO resolves non-uniqueness along Jacobian-null (gauge) directions, but it does not directly cure ill-conditioning in function-relevant directions associated with small yet nonzero singular values.

(ii) The method introduces a dissipation/time-scale parameter controlling how aggressively the history variable tracks the instantaneous Dirac-Frenkel reference. In this work we use simple, problem-dependent choices; principled and automated selection strategies (e.g., based on conditioning indicators or residual) remain to be developed.

## Acknowledgments

This work was partially supported by the National Science Foundation grant 2046521 and by the Office of Naval Research award N00014-22-1-2728.

## Impact Statement

This paper presents work whose goal is to advance the field of Machine Learning. There are many potential societal consequences of our work, none of which we feel must be specifically highlighted here.

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

## Appendix Outline

This appendix complements the main text as follows.

- Section A provides details on the two examples of Section 2.2 (wave collision and advection–reaction) illustrating tangent-space collapse and how DFO escapes it.

- Section B collects the specifications for the PDE benchmarks of Sections 5.2 and 5.3.

- Section C reports details about architectural choices, least-squares solvers, IC fitting, some extended numerical results, and baselines for the experiments in Sections 5.2 and 5.3.

- Section D summarizes the randomized tSVD variant used in RDFO (Table 2) to reduce the cost of least-squares and to approximate nullspace projections.

- Section E specifies the RK4 time discretization used for DFO in the PDE experiments of Section 5.2.

## A. Examples with Tangent-Space Collapse

### A.1. Wave collision problem

We consider the 1D wave equation as in the main text $\partial_t^2 u = c^2 \partial_x^2 u$ with initial conditions $u(0, x) = \phi_0(x; -2) + \phi_\rho(x; 2)$ and $u_t(0, x) = c\partial_\mu \phi_0(x; -2) - c\partial_\mu \phi_\rho(x; 2)$, where we recall $\phi_\rho(x; \mu) = \exp(-\frac{1}{2}(x - \mu)^2/(1 + \rho))$ for a parameter $\rho \geq 0$. It can be easily checked that the analytical solution is given by

$$u(t, x) = \phi_0(x - ct; -2) + \phi_\rho(x + ct; 2). \tag{18}$$

We bring the equation in first-order form and use the two-wave parameterization $\hat{u}(\theta(t), x)$ from Section 2.2, see (6). We let $c = 1$ and study the case $\rho = 0$, where the tangent space loses rank at collision. Indeed, at a collision event the two wave locations coincide, namely $\theta_1 = \theta_2$, and $\partial_{\theta_1}\hat{u}(\theta, \cdot) \equiv \partial_{\theta_2}\hat{u}(\theta, \cdot)$, so that the direction

$$\xi_1 = [1, -1, 0, 0]^\top \in \text{null}(G(\theta)). \tag{19}$$

If additionally $\theta_3 = \theta_4$, then $\partial_{\theta_3}\hat{u}(\theta, \cdot) \equiv -\partial_{\theta_4}\hat{u}(\theta, \cdot)$, so also

$$\xi_2 = [0, 0, 1, 1]^\top \in \text{null}(G(\theta)). \tag{20}$$

Consequently, the tangent space dimension drops and the DF least-squares admits infinitely many parameter velocities. Let $\bar{\eta}(\theta(t))$ denote the DF reference velocity (minimal-norm solution). By construction, $\bar{\eta}(\theta(t)) \in \text{range}(G(\theta(t)))$ and therefore $\bar{\eta}(\theta(t)) \perp \text{null}(G(\theta(t)))$, hence

$$\xi_1^\top \bar{\eta}(\theta(t)) = 0 \quad \text{and} \quad \xi_2^\top \bar{\eta}(\theta(t)) = 0 \qquad \text{whenever } \xi_1, \xi_2 \in \text{null}(G(\theta(t))). \tag{21}$$

Thus, at collision the minimal-norm DF velocity has no component along the critical separation direction $\xi_1$.

**Why DFO escapes at collapse.** Let now $\xi = \xi_1$ and define the scalar components

$$\beta(t) = \xi^\top \bar{\eta}(\theta(t)), \qquad \alpha(t) = \xi^\top m(t).$$

The Onsager equation $\tau \dot{m} = \bar{\eta} - m$ implies $\tau \dot{\alpha}(t) = \beta(t) - \alpha(t)$. Before collision, $\text{null}(G(\theta(t))) = \{0\}$ and thus $P(\theta(t)) = 0$; therefore DFO reduces to DF at the level of $\dot{\theta}$, but $m(t)$ still accumulates a smooth history of $\bar{\eta}(\theta(t))$, so $\alpha(t)$ is generically nonzero near collision. At the collision time (collapse), $\xi \in \text{null}(G(\theta(t)))$ and therefore $\xi^\top P(\theta(t))m(t) = \xi^\top m(t) = \alpha(t)$. Using the DFO gauge-fixed update $\dot{\theta} = \bar{\eta} + \lambda P(\theta)m$, we obtain

$$\xi^\top \dot{\theta}(t) = \xi^\top \bar{\eta}(\theta(t)) + \lambda \xi^\top P(\theta(t))m(t) = 0 + \lambda \alpha(t) \neq 0,$$

so DFO produces a nonzero velocity component along $\xi$ and the waves can separate after collision. Furthermore, we can make the following exact-recovery statement for a specific choice of the DFO parameter.

**Proposition A.1** (DFO recovers the exact crossing continuation). *Consider the collapse case $\rho = 0$ and $c = 1$. Let*

$$\theta^*(t) = [-2 + t,\, 2 - t,\, -2 + t,\, 2 - t]^\top, \qquad q = \dot{\theta}^*(t) = [1,\, -1,\, 1,\, -1]^\top. \tag{22}$$

*Assume that the continuous-time DFO dynamics $\tau \dot{m} = \bar{\eta}(\theta) - m$ and $\dot{\theta} = \bar{\eta}(\theta) + \lambda P(\theta)m$ are initialized with $\theta(0) = \theta^*(0)$ and $m(0) = 0$, and choose*

$$\lambda = (1 - \exp(-2/\tau))^{-1}. \tag{23}$$

*Then the path $\theta^*(t)$ satisfies the continuous-time DFO dynamics and the represented function $\hat{u}(\theta^*(t), \cdot)$ equals the exact wave-equation solution (18) for all $t \geq 0$.*

*Proof.* Along $\theta^*(t)$, the parametrization (6) represents the exact solution (18). For $t \neq 2$, the path $\theta^*(t)$ avoids the rank-loss configuration described by (19)–(20), and the exact PDE vector field satisfies

$$\mathcal{F}(\hat{u}(\theta^*(t), \cdot)) = \nabla_\theta \hat{u}(\theta^*(t), \cdot)^\top q. \tag{24}$$

By (24), for $t \neq 2$, the minimal-norm DF reference velocity is unique and satisfies $\bar{\eta}(\theta^*(t)) = q$. Before collision, the same rank condition from (22) and (19)–(20) gives $P(\theta^*(t)) = 0$, so DFO agrees with DF at the level of $\dot{\theta}$. Since $m(0) = 0$ and $\bar{\eta}(\theta^*(t)) = q$ for $0 \leq t < 2$, the Onsager equation gives

$$m(t) = (1 - \exp(-t/\tau))\, q, \qquad 0 \leq t < 2. \tag{25}$$

Taking the left limit in (25) at collision yields

$$m(2) = (1 - \exp(-2/\tau))\, q. \tag{26}$$

At $t = 2$, the path (22) satisfies $\theta_1^* = \theta_2^*$ and $\theta_3^* = \theta_4^*$, so the collision nullspace directions are those in (19) and (20). The exact velocity decomposes as

$$q = [0, 0, 1, -1]^\top + \xi_1, \tag{27}$$

where $[0, 0, 1, -1]^\top$ is orthogonal to the collapse directions $\xi_1$ and $\xi_2$ from (19)–(20). Therefore, by (27) and the minimal-norm orthogonality relation (21), the DF reference velocity at collision is

$$\bar{\eta}(\theta^*(2)) = [0, 0, 1, -1]^\top,$$

and, by (26) and (27), the projected memory in the DFO correction is

$$P(\theta^*(2))m(2) = (1 - \exp(-2/\tau))\, \xi_1.$$

Hence, with the choice in (23), the DFO velocity at collision is

$$\dot{\theta}(2) = \bar{\eta}(\theta^*(2)) + \lambda P(\theta^*(2))m(2) = [0, 0, 1, -1]^\top + \xi_1 = q, \tag{28}$$

and the separation component is restored exactly.

After collision, $\theta^*$ immediately leaves the rank-loss configuration described, so the unique DF reference velocity along the exact path is again $q$. Hence, $P(\theta^*(t)) = 0$ and $\dot{\theta}(t) = q$ for $t > 2$. Together with (24) and (28), this shows that $\theta(t) = \theta^*(t)$ is a solution of the DFO dynamics, and the claim is proven. $\square$

**Numerical setup to produce Figure 1.** For the numerics, we consider two regimes: the *collapse case* $\rho = 0$ and an *ill-conditioned case* $\rho = 4.635 \times 10^{-6}$. The Onsager parameters are set to $\tau = 0.5$ and $\lambda = 1$. We assemble the batch Jacobian $J(\theta)$ and right-hand side $f(\theta)$ using $N = 151$ equidistant collocation points on the periodic domain $\Omega = [-12, 12)$. For this toy parameterization, the spatial derivatives entering the wave operator are evaluated analytically. We discretize both DF and DFO in time with forward Euler and the semi-implicit scheme of Section 4.3, respectively, using step size $\delta t = 3 \times 10^{-4}$, and at each time step compute the DF reference velocity $\bar{\eta}_k$ via a tSVD-based least-squares solve of the batch system.

## A.2. Advection–reaction problem

Consider the advection–reaction equation on $\Omega = [0, 2\pi)$ with periodic boundary conditions

$$\partial_t u(t, x) = -c\,\partial_x u(t, x) - \kappa\,u(t, x) + s(t, x),$$

with $c \in \mathbb{R}$ and $\kappa \in \mathbb{R} \setminus \{0\}$. We choose $c = 1$ and $\kappa = 1$. We select the initial condition and source term $s(t, x)$ so that the exact solution is

$$u(t, x) = a(t)\sin(x) + b(t)\cos(x), \qquad a(t) = \sin(t), \quad b(t) = \sin(t),$$

so in particular $\partial_t u(\pi/2, \cdot) \equiv 0$. A direct calculation shows that the choice

$$s(t, x) = \big(\cos(t) + (\kappa - c)\sin(t)\big)\sin(x) + \big(\cos(t) + (\kappa + c)\sin(t)\big)\cos(x)$$

makes the above $u(t, x)$ satisfy the PDE exactly. We approximate $u$ with the two-parameter ansatz

$$\hat{u}(\theta(t), x) = \hat{a}(\theta(t))\sin(x) + \hat{b}(\theta(t))\cos(x), \qquad \hat{a}(\theta) = \sin(\theta_1),\ \hat{b}(\theta) = \sin(\theta_2),$$

with $\theta = [\theta_1, \theta_2]^\top \in \mathbb{R}^2$. The parameter derivatives are

$$\partial_{\theta_1}\hat{u}(\theta, x) = \cos(\theta_1)\sin(x), \qquad \partial_{\theta_2}\hat{u}(\theta, x) = \cos(\theta_2)\cos(x),$$

hence the tangent space collapses whenever $\cos(\theta_1) = 0$ or $\cos(\theta_2) = 0$, e.g. at $\theta_i = \pi/2$ modulo $2\pi$. In particular, the chosen parameterization matches the solution exactly with $\theta_1(t) = \theta_2(t) = t$, which can be obtained from the DF least-squares problem at times $t \in [0, \pi/2]$ as the Jacobian is full rank. When $t$ hits $\pi/2$, we have that $\cos(\theta_1(t)) = \cos(\theta_2(t)) = 0$, and both derivatives $\partial_{\theta_1}\hat{u}$ and $\partial_{\theta_2}\hat{u}$ vanish and the tangent space collapses to $\{0\}$. Hence, the DF least-squares admits every velocity as a minimizer, and the minimal-norm selection returns $\bar{\eta}(\theta(t)) = 0$, and the DF trajectory freezes at the collapse point once it reaches it (see Figure 2b).

**Why DFO escapes.** At a collapse point, the Gram matrix satisfies $G(\theta) = 0$ and thus the nullspace projector is $P(\theta) = I$. Moreover $\bar{\eta}(\theta(t)) = 0$ at $t = \pi/2$. Since $\bar{\eta}(\theta(t)) \neq 0$ away from collapse, the Onsager filter produces a nonzero history $m(\pi/2) \neq 0$. Therefore DFO yields

$$\dot{\theta}(\pi/2) = \bar{\eta}(\pi/2) + \lambda P(\theta(\pi/2))m(\pi/2) = \lambda m(\pi/2) \neq 0,$$

so the parameter trajectory leaves the collapse set immediately and the DF evolution resumes uniquely once $G(\theta)$ regains rank.

**Numerical setup to produce Figure 2.** We use an equidistant spatial grid with $N = 512$ points on $[0, 2\pi)$ and integrate over the time interval $[0, 6]$ with step size $\delta t = 10^{-4}$. For DF and DFO, we discretize the parameter dynamics for $\theta$ with forward Euler, while the history variable $m$ is advanced with backward Euler, resulting in the semi-implicit scheme (16) from the main text. The SVD-based least-squares solver employs truncation thresholds given by an absolute tolerance $10^{-3}$ and a relative tolerance $10^{-10}$. Finally, we set the Onsager parameters to $\tau = 0.05$ and $\lambda = 10^{-5}$.

# B. PDE Examples Specifications

Let us provide a detailed description of the PDEs we use for the experiments presented in Sections 5.2 and 5.3. We show the reference (truth) and the solution obtained with the DFO scheme for the 4 PDEs in Figures 6 to 9.

## B.1. Rotating detonation waves (RDW)

We consider the rotating detonation wave model of Koch et al. (2020), motivated by rotating detonation engines for space propulsion. The state consists of two fields on the periodic domain $\Omega = [0, 2\pi)$: an intensive fluid property $\eta(t, x)$ and the combustion progress $\lambda(t, x)$. The dynamics generate a sharp-fronted wave that travels around the circular domain and, through the reaction–relaxation coupling, can trigger a secondary wave. Specifically, we solve the following two-field system with periodic boundary conditions,

$$\partial_t\eta = -\eta\,\partial_x\eta + \nu\,\partial_{xx}\eta + (1 - \lambda)\,\omega(\eta) + \xi(\eta), \qquad \partial_t\lambda = \nu\,\partial_{xx}\lambda + (1 - \lambda)\,\omega(\eta) - \beta(\eta; \mu)\,\lambda,$$

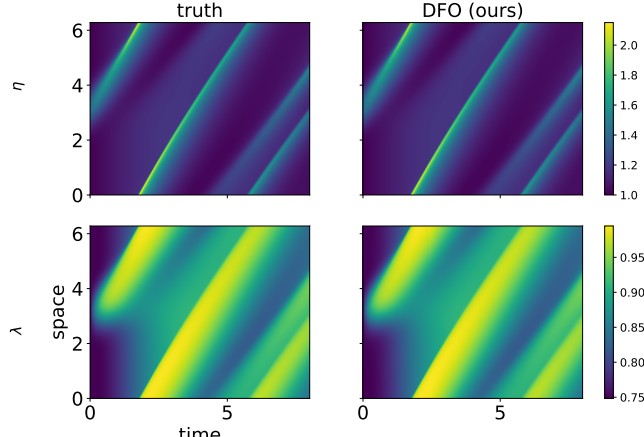

*Figure 6.* Space-time plot of the fields $\eta$ and $\lambda$, solutions of the rotating detonation waves PDE system.

where

$$\omega(\eta) = \exp\left(\frac{\eta - \eta_c}{\alpha}\right), \quad \beta(\eta; \mu) = \frac{\mu}{1 + \exp(r(\eta - \eta_p))}, \quad \xi(\eta) = -\varepsilon\,\eta.$$

We use the parameters $\nu = 10^{-2}$, $\mu = 3.5$, $\alpha = 0.3$, $\eta_c = 1.1$, $\varepsilon = 0.11$, $\eta_p = 0.5$, $r = 5.0$, and the initial condition

$$\eta(0, x) = 0.4 \exp\left(-2.25(x - \pi)^2\right) + 1, \qquad \lambda(0, x) = 0.75.$$

We integrate until final time $T = 8$. Since no closed-form solution is available, we compute reference solutions using RK4 in time coupled with a high-order IMEX discretization in space.

### B.2. Transport through flow field

We model passive transport through a nonuniform flow field via a two-dimensional linear advection equation. The unknown $u(t, x, y)$ represents a scalar quantity (e.g., a tracer concentration) that is carried by a spatially varying velocity field on the periodic domain $\Omega = [-1, 1)^2$. Specifically, we solve

$$\partial_t u + c_x(x)\,\partial_x u + c_y(y)\,\partial_y u = 0, \qquad (x, y) \in [-1, 1)^2,$$

with periodic boundary conditions and separable advection speeds

$$c_x(x) = 1.0\left(1 + 0.6\sin\left(\tfrac{2\pi \cdot 3(x - x_0)}{L_x}\right)\right), \qquad c_y(y) = 0.8\left(1 + 0.3\cos\left(\tfrac{2\pi \cdot 2(y - y_0)}{L_y}\right)\right), \qquad L_x = L_y = 2.$$

We initialize with a localized Gaussian bump,

$$u(0, x, y) = \exp\left(-\frac{(x - \bar{x})^2 + (y - \bar{y})^2}{\pi\sigma}\right), \quad \sigma = 8 \times 10^{-3},\ \bar{x} = -0.2,\ \bar{y} = 0,$$

and integrate until final time $T = 20$. Since no closed-form solution is available for this spatially varying velocity field, we compute reference solutions using RK4 in time combined with fourth-order finite differences in space.

### B.3. Charged particles in electric field

Following a setup similar to (Berman & Peherstorfer, 2024), we consider a two-dimensional (one space, one velocity) Vlasov model for collisionless charged particles in a prescribed electrostatic potential. The unknown $u(t, x, v)$ is the particle density in phase space; starting from a localized distribution, the self-consistent transport in $(x, v)$ driven by free streaming and electric forcing advects, shears, and deforms the density over time. Specifically, we solve the Vlasov equation on $[-1, 1)^2$,

$$\partial_t u(t, x, v) = -v\,\partial_x u(t, x, v) + (\partial_x \phi(x))\,\partial_v u(t, x, v), \qquad (x, v) \in [-1, 1)^2,$$

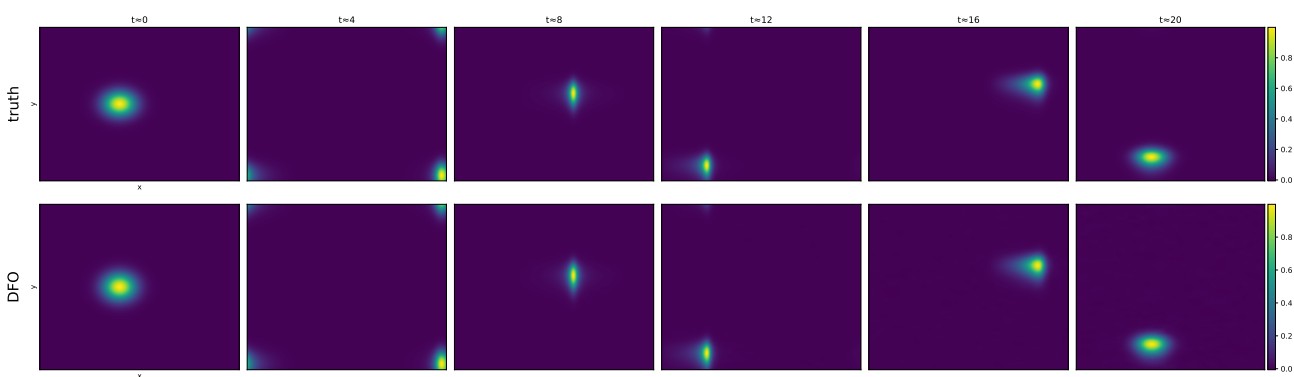

*Figure 7.* Snapshots of the solution of the advection equation modeling transport through flow field

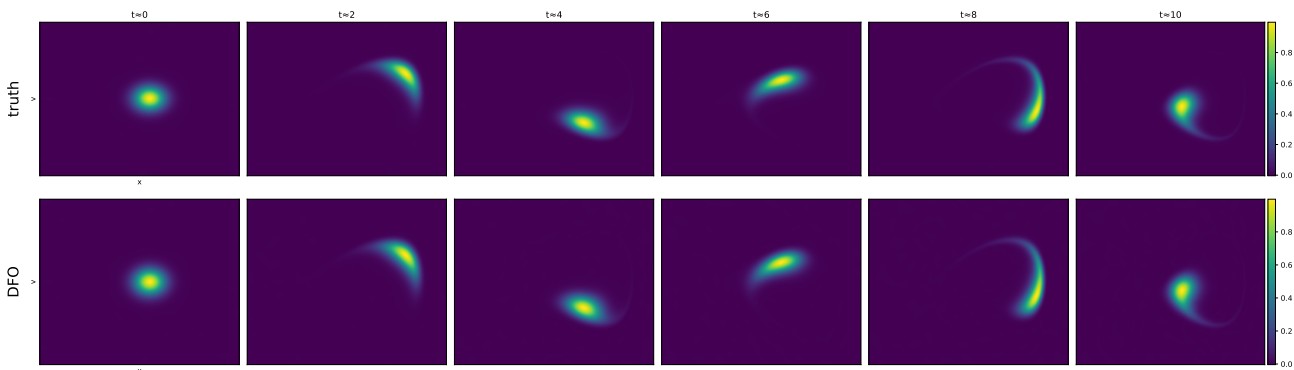

*Figure 8.* Snapshots of the charged particles density solution of the Vlasov equation

with periodic boundary conditions in both position $x$ and velocity $v$. The potential is taken as

$$\phi(x) = -\alpha \left(1 + \cos(\pi(x + \mu)^4)\right) - \beta \sin(\pi x),$$

and we set $\alpha = 0.2$, $\beta = 0.1$, $\mu = 0$. The initial condition is a localized Gaussian bump in phase space,

$$u(0, x, v) = \exp\left(-\frac{(x - 0.1)^2 + v^2}{\sigma^2}\right), \qquad \sigma = 0.15,$$

and we integrate until final time $T = 10$. Since no closed-form solution is available, we compute reference solutions using RK4 in time coupled with fourth-order finite differences in space.

### B.4. Fokker–Planck equation

We adopt the setup of (Bruna et al., 2024), where the dynamics are described in terms of $d$ interacting particles $X(t) = (X_1(t), \ldots, X_d(t))^\top \in \mathbb{R}^d$ evolving in an anharmonic confining potential with an additional repulsive interaction term. Specifically, for $i = 1, \ldots, d$,

$$\mathrm{d}X_i(t) = \left(g(t, X_i(t)) + \alpha \bar{X}(t) - \alpha X_i(t)\right) \mathrm{d}t + \sqrt{2D} \, \mathrm{d}W_i(t), \qquad \bar{X}(t) = \frac{1}{d} \sum_{j=1}^d X_j(t),$$

where $W_1, \ldots, W_d$ are independent Wiener processes. We start from a Gaussian initial law $X(0) \sim \mathcal{N}(m_0, \Sigma_0)$ (equivalently, $u(0, \cdot)$ is Gaussian), but due to the nonlinearity the density does not remain Gaussian over time. The joint probability density $u(t, x)$ of $X(t)$, with $x = (x_1, \ldots, x_d)^\top \in \mathbb{R}^d$, satisfies the $d$-dimensional Fokker–Planck equation

$$\partial_t u = -\sum_{i=1}^d \partial_{x_i}\big(u \, h_i(t, x)\big) + D \sum_{i=1}^d \partial_{x_i}^2 u, \qquad h_i(t, x) = g(t, x_i) + \alpha \bar{x} - \alpha x_i, \qquad \bar{x} = \frac{1}{d} \sum_{j=1}^d x_j,$$

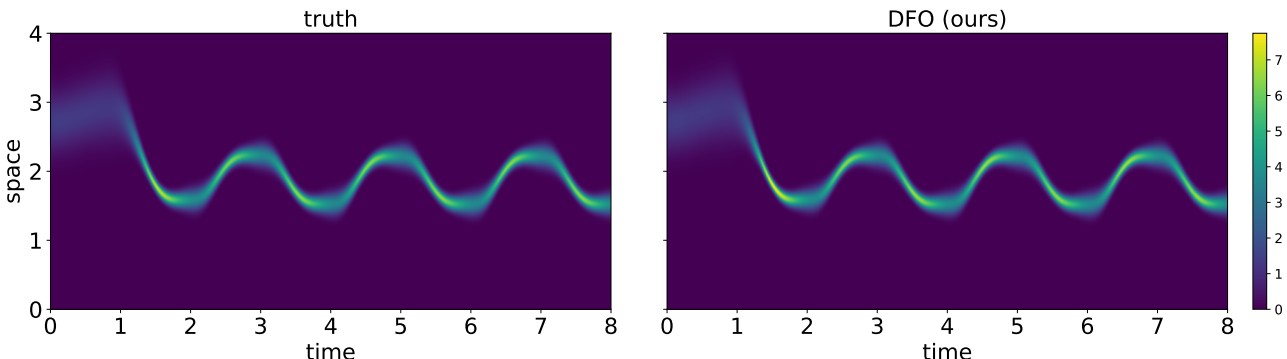

*Figure 9.* Space-time plot of the marginal density of the fifth component of the solution of the Fokker-Planck equation

with

$$g(t, x) = (a(t) - x)^3, \qquad a(t) = a_{\text{scale}}\big(\sin(\pi a_{\text{freq}} t) + a_{\text{shift}}\big).$$

We solve the equation in $d = 5$ dimensions, and set the parameters to $D = 10^{-2}$, $\alpha = -0.5$, $a_{\text{scale}} = 1.25$, $a_{\text{shift}} = 1.5$, $a_{\text{freq}} = 1$. The final time is $T = 8$. To compute reference statistics, we simulate the SDE with an Euler–Maruyama scheme to generate $10^5$ i.i.d. particle trajectories and form the empirical mean and covariance.

## C. Detailed Numerical Setups

### C.1. Architectures and periodic embedding

All experiments use MLP parametrizations equipped with a periodic embedding layer so that periodic boundary conditions are satisfied by construction. Concretely, we use the periodic embedding from (Berman & Peherstorfer, 2023): for an input $x \in \mathbb{R}^d$ with period $P$, each embedding channel is defined by trainable parameters $a, \phi, b \in \mathbb{R}^d$ as

$$\texttt{periodic\_embed}(x) \;=\; \sum_{i=1}^{d} \big[a \cos((2\pi/P)\, x + \phi) + b\big]_i,$$

and this operation is repeated $w$ times (with independent parameters $(a, \phi, b)$ per channel) to produce an output vector $y \in \mathbb{R}^w$. For the low-dimensional examples (RDW, linear advection, and Vlasov), we fix $a \equiv \mathbf{1}$ and $b \equiv \mathbf{0}$ in the embedding, i.e., only the phase parameters $\phi$ are optimized. For RDW, we use two separate MLPs (one for $\eta$ and one for $\lambda$), each with 4 layers of width 10, for a total of 922 trainable parameters. For Vlasov and 2D advection, we use a single MLP with 3 layers of width 32 (3265 parameters). For the 5D Fokker–Planck equation, we use a single MLP with 3 layers of width 20 (1581 parameters) and parameterize the density as $\hat{u}(\theta, x) = \exp(-\text{NN}_\theta(x))$. All networks use the swish activation function.

### C.2. Solving the least-squares subproblem

The core procedure at each time step of all schemes based on the Dirac-Frenkel residual minimization principle is the solution of the linear least-squares problem

$$\min_{\eta \in \mathbb{R}^p} \; \|J\eta - f\|_2^2, \tag{29}$$

where $J$ and $f$ are as defined in (17), possibly regularized to handle ill-conditioning. Let us review two standard forms of regularization: (i) a truncated SVD (tSVD), and (ii) explicit Tikhonov regularization.

**tSVD regularization.** Let $J = U\Sigma V^\top$ be the SVD, with singular values $\sigma_1 \geq \cdots \geq \sigma_p \geq 0$. Given a truncation level $\epsilon > 0$, which in all our experiments using the tSVD subroutine we set relative to the largest singular value $\epsilon = \epsilon_{\text{rel}}\sigma_1$, we retain only the modes with $\sigma_i \geq \epsilon$, and define the truncated pseudoinverse

$$J_\epsilon^\dagger = V\Sigma_\epsilon^\dagger U^\top, \qquad (\Sigma_\epsilon^\dagger)_{ii} = \begin{cases} \sigma_i^{-1}, & \sigma_i \geq \epsilon, \\ 0, & \sigma_i < \epsilon. \end{cases}$$

The resulting regularized minimal-norm solution to (29) is $\eta = J_\epsilon^\dagger f$.

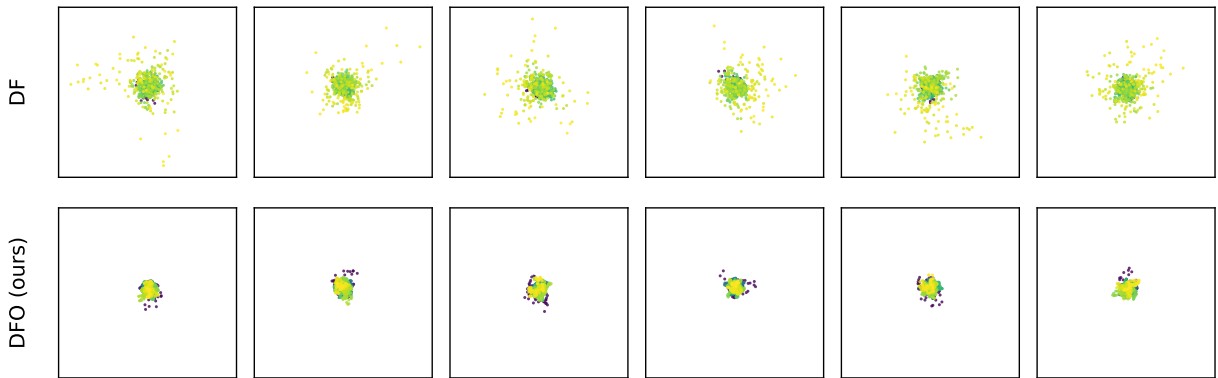

*Figure 10.* Vlasov experiment: six random two-dimensional projections of the parameter velocities (which live in $\mathbb{R}^{3265}$) produced by tSVD-regularized DF (top row) and by DFO (bottom row). Each panel corresponds to an independent random projection. Markers are colored with a time gradient, with lighter dots indicating later times. DF yields more erratic projected velocities with intermittent large excursions, whereas DFO produces more coherent trajectories in these projections, consistent with the discussion in the main text.

**Tikhonov regularization.** Alternatively, one can solve the ridge-regularized problem

$$\min_{\eta \in \mathbb{R}^p} \; \|J\eta - f\|_2^2 + \gamma \|\eta\|_2^2, \qquad \gamma > 0,$$

whose solution is $\eta = (J^\top J + \gamma I)^{-1} J^\top f$.

**Numerical implementation (QR + SVD).** In both cases we avoid forming the full SVD of $J$ by computing a thin QR factorization $J = QR$ followed by an SVD of the smaller triangular factor $R = \widetilde{U} \Sigma V^\top$. Since $J = (Q\widetilde{U})\Sigma V^\top$, the singular values are those of $R$, and the right singular vectors are $V$. We then apply either (i) hard truncation where $\sigma_i < \epsilon$ is set to 0 (tSVD), or (ii) the Tikhonov filter $\sigma_i \mapsto \sigma_i^2/(\sigma_i^2 + \gamma)$ to obtain the regularized least-squares solution. The cost of this procedure scales in both cases like $O(Np^2)$ when $N \geq p$, which is the case for all our experiments.

### C.3. Time and space discretization

For the low-dimensional examples in Section 5.2, we integrate the parameter dynamics for $\theta$ with a fixed-step RK4 scheme across all methods, except for TENG, which uses the Heun method as suggested in (Chen et al., 2024). In all cases, the auxiliary momentum variable $m$ is advanced via a stage-coupled EMA (see Section E). We use a uniform time step $\delta t = 4 \times 10^{-3}$ for every method. Spatial discretizations rely on equispaced grids with: RDW $N_x = 2048$, Vlasov $N_x = N_v = 500$, and 2D advection $N_x = N_y = 512$.

For the Fokker–Planck example, we use explicit Euler in time for all methods (including TENG), while DFO employs the specialized Euler discretization described in Section 4.3 of the main text. We truncate $\mathbb{R}^5$ to a periodic box $x \in [0,4]^5$ and adopt an adaptive sampling strategy: at each time step we draw a mixture of $2 \times 10^3$ uniform points and $2 \times 10^3$ Gaussian points, where the Gaussian component is fitted to the current solution using samples from the previous step.

### C.4. Vlasov: temporal regularity of parameter velocities

As noted in the main text, the DFO correction promotes parameter dynamics that are smoother and less erratic in time than ones obtained from the tSVD-regularized DF scheme. To make this effect visible in the high-dimensional parameter space of the Vlasov experiment, we visualize the (discrete) parameter velocities produced by DF and DFO over the full time integration through several independent random two-dimensional projections.

Figure 10 shows that DF produces highly scattered projected velocities with intermittent large excursions, consistent with erratic changes in the effective update direction when the batch Jacobian is close to rank-deficient and the active tSVD subspace changes over time. In contrast, DFO yields projected velocities that remain significantly more concentrated and coherent across time, highlighting the temporally smoothing effect of the history variable and its nullspace-projected correction.

## C.5. Benchmarks

We run NIVP with Tikhonov regularization and use the direct least-squares solver described in Appendix C.2 to keep the procedure simple; we use 10 restarts with $50k$ Adam steps as suggested in (Finzi et al., 2023). In the Fokker–Planck experiment, the refit objective is chosen to match the IC-fit objective, and we use a truncated SVD in place of Tikhonov regularization, as the latter is ineffective in this setting.

For RSNG, we adopt the sparse sketching strategy (column sampling) proposed in the original paper as this is a core part of the method which can improve accuracy by preventing overfitting (Berman & Peherstorfer, 2023). We report results averaged over 5 random seeds in Tables 1 and 2.

For TENG, we use the randomized variant to solve the least-squares subproblems with 7 iterations for the first stage (Euler and Heun) and 5 iterations for the second stage (Heun), as suggested in (Chen et al., 2024).

## C.6. Fitting the initial condition

For RDW we train with Adam for $50,000$ iterations, while for Vlasov, linear advection, and Fokker–Planck we use Adam for $100,000$ iterations, all using the mean-squared loss. In the Fokker–Planck case, we fit $\log(u_0)$ using a mixture of uniform samples on the periodic box and Gaussian samples with importance weighting to define the loss function.

## C.7. Additional numerical experiments

We add two robustness checks for the five-dimensional Fokker–Planck experiment. First, we vary the relative tSVD truncation threshold $\epsilon_{\mathrm{rel}}$. This threshold is a numerical resolution parameter for both DF and DFO: singular directions below it are not treated as stably resolved by the least-squares solve. The relevant comparison is therefore whether DFO improves over DF over a reasonable range of truncation levels. Table 3 shows that tuned DFO gives lower error of the mean for all tested values $\epsilon_{\mathrm{rel}} \in [10^{-5}, 10^{-2}]$.

*Table 3.* Fokker–Planck robustness sweep over the relative tSVD truncation threshold. Entries are relative errors of the mean, averaged over five runs.

| $\epsilon_{\mathrm{rel}}$ | DF + tSVD | DFO |
| --- | --- | --- |
| 1.00e−5 | 7.75e−2 | 6.26e−2 |
| 5.00e−5 | 4.73e−2 | 3.99e−3 |
| 1.00e−4 | 1.46e−2 | 4.54e−3 |
| 1.00e−3 | 1.11e−2 | 6.60e−3 |
| 1.00e−2 | 2.46e−2 | 1.88e−2 |

Second, we vary the Onsager parameter $\lambda$, which scales the projected momentum correction, and keep $\beta = 0.2$ fixed. Since this correction is added to the DF reference velocity, we use order-one values and tune only within a modest range. Table 4 shows that the Fokker–Planck mean error is stable for $\lambda \in [0.5, 2.5]$, so the method does not appear to require fine tuning of $\lambda$ in this example.

*Table 4.* Fokker–Planck robustness sweep over the Onsager parameter $\lambda$. Entries are relative errors of the mean, averaged over five runs.

| $\lambda$ | DFO |
| --- | --- |
| 0.5 | 1.36e−2 |
| 0.75 | 1.53e−2 |
| 1 | 2.20e−2 |
| 1.25 | 1.65e−2 |
| 1.50 | 7.76e−3 |
| 1.75 | 1.40e−2 |
| 2.00 | 8.53e−3 |
| 2.25 | 3.51e−2 |
| 2.50 | 7.75e−3 |

## D. Randomization

When the batch Jacobian $J(\theta) \in \mathbb{R}^{N \times p}$ has rapidly decaying spectrum, we can approximate its dominant right singular subspace via a randomized SVD to reduce costs (Halko et al., 2011).

**Randomized SVD (RSVD).** Let $\Gamma \in \mathbb{R}^{p \times s}$ be a sketching matrix (e.g. Gaussian, with oversampling as needed). Form

$$Y = J(\theta)\Gamma \in \mathbb{R}^{N \times s}, \qquad Q = \text{orth}(Y) \in \mathbb{R}^{N \times s}.$$

Then compute the (small) SVD of the compressed matrix $B = Q^\top J(\theta) \in \mathbb{R}^{s \times p}$,

$$B = \tilde{U}\Sigma V^\top,$$

which yields an approximate right singular basis $V$. After truncation at tolerance $\epsilon$ (tRSVD), we obtain $V_\epsilon$ and define the approximate nullspace projector

$$\tilde{P}_\epsilon(\theta)z = z - V_\epsilon(V_\epsilon^\top z).$$

The computational complexity of this scheme scales like $O(Ns^2 + ps^2)$, which amounts to $O(Ns^2)$ in our numerical experiments as $N \geq p$. Plugging tRSVD into Algorithm 1 in place of the full tSVD computation yields the randomized version of the DFO scheme (RDFO) shown in Table 2.

## E. RK4 Time Discretization of DFO

This section describes the RK4 time discretization of the DFO dynamics used in the low-dimensional PDE experiments in Section 5.2. Let the macro step size be $h$ and the current state be $(\theta_k, m_k)$. Define the RK4 stage substeps $h_1 = h_4 = h$ and $h_2 = h_3 = h/2$, and the intermediate states

$$\theta_1 = \theta_k, \qquad \theta_2 = \theta_k + \tfrac{h}{2}k_1, \qquad \theta_3 = \theta_k + \tfrac{h}{2}k_2, \qquad \theta_4 = \theta_k + h\,k_3.$$

At each stage $s = 1, 2, 3, 4$, we evaluate $J_s = J(\theta_s)$ and $f_s = f(\theta_s)$, and solve the least-squares subproblem using a tSVD with tolerance $\epsilon\,\sigma_{\max}(J_s)$ (see Section C). This yields (i) the minimal-norm Dirac-Frenkel velocity $\bar{\eta}_s$, and (ii) the retained right-singular basis $V_s$ associated with singular values above the truncation threshold. Given $(\theta_s, m_{s-1})$, we then perform the stage updates

$$\Delta_{\text{ls},s} = h_s\,\bar{\eta}_s, \qquad m_s = \beta\,m_{s-1} + (1-\beta)\Delta_{\text{ls},s}, \qquad m_s^\perp = m_s - V_s(V_s^\top m_s),$$

and define the stage direction

$$k_s = \bar{\eta}_s + \lambda\frac{m_s^\perp}{h_s} = \bar{\eta}_s + \lambda\frac{m_s - V_s(V_s^\top m_s)}{h_s}.$$

Here $\beta \in (0, 1)$ controls the history (EMA) update and $\lambda \geq 0$ scales the orthogonal momentum correction. Finally, the RK4 update is

$$\theta_{k+1} = \theta_k + \frac{h}{6}\big(k_1 + 2k_2 + 2k_3 + k_4\big), \qquad m_{k+1} = m_4.$$

