# OpenReview forum: "A Dirac-Frenkel-Onsager Principle: Instantaneous Residual Minimization with Gauge Momentum for Nonlinear Parametrizations of PDE Solutions"
_ICML.cc/2026/Conference — ICML 2026 spotlight_

### Official Review · Reviewer_bzX3 · 2026-03-10

**Soundness:** 3
**Presentation:** 3
**Significance:** 3
**Originality:** 3
**Overall Recommendation:** 5
**Confidence:** 4

**Summary:**

This work considers solving PDEs involving time derivatives by parameterizing the solution at each point in time using a neural networks, and then learning the trajectory of the neural network parameter. This is done by deriving an ODE which projects to the true time gradient onto the tangent space at the current parameter values. One major issue with this approach is that the map from parameters to the tangent space may have a kernel, in this case the correct direction in parameter space is not uniquely determined by the equation. It is shown by example that this leads to incorrect dynamics on certain problem, such as the wave equation. The authors propose to address this issue by introducing a selection scheme that uses the trajectory history to determine a null space vector to inject at each time step. Experimentally, the method demonstrates good performance.

**Compliance With Llm Reviewing Policy:**

Affirmed.

**Key Questions For Authors:**

Where does Onsager's minimal dissipation principle come from? Is it possible to show (under certain assumptions or for certain PDEs) that the proposed approach can always find the correct PDE trajectory, as demonstrated experimentally?

**Limitations:**

yes

**Strengths And Weaknesses:**

Strengths:
- The paper is well-written and easy to understand.
- The problem addressed is clearly explained and the authors provide an example demonstrating the issue at hand on the wave equation.
- The suggested solution is simple and easy to implement.
- The experimental results show that the method performs as intended and alleviates the issues encountered with the previous norm-minimization approach.

Weaknesses:
- The proposed method is rather ad hoc. In particular, the authors intuitively motivate the method, but there is no theory showing that the correct PDE solution can always be found using their method.
- The method does introduce an additional hyperparameter which must be tuned. The authors do discuss this in the paper, however.

---

> ### Author Rebuttal · Authors · 2026-03-30
>
> Thank you for the careful reading and for recognizing both the clarity of the presentation and the empirical effectiveness of the method.
>
> *"The proposed method is rather ad hoc. [...] Where does Onsager's minimal dissipation principle come from? "*
> - While the approach might look ad-hoc at first, we stress that our DFO preserves the instantaneous function-space residual minimizer of Dirac-Frenkel. In this sense, we maintain the principled foundation of Dirac-Frenkel residual minimization. Furthermore, the Onsager construction is a principled variational rule for describing an evolution law that represents a temporally coherent, low-pass filtered time-dependent quantity (in our case, the reference velocity). In that sense, the Onsager balances fidelity to the current Dirac-Frenkel velocity with resistance to abrupt gauge changes.
>
>
> *"In particular, the authors intuitively motivate the method, but there is no theory showing that the correct PDE solution can always be found using their method. Is it possible to show (under certain assumptions or for certain PDEs) that the proposed approach can always find the correct PDE trajectory, as demonstrated experimentally?"*
>
> - Indeed, for specific PDEs and parametrizations, we can show that DFO recovers the exact trajectory whereas DF incurs an error of $O(1)$. Let us take the colliding-wave example. The exact (strong) PDE solution lies exactly in the trial manifold with parameter path $\theta^\ast(t) = [-2 + t, 2 - t, -2 + t, 2 - t]^T$. Away from the collision time, the Jacobian is full rank and so DF and DFO follow exactly this path. At collision ($t = 2$), the minimal-norm DF gets stuck because an update is needed in the nullspace direction $[1, -1, 0, 0]^T$ (shown in the paper). In contrast, DFO injects exactly this missing component through $P(\theta)m(t)$ if $m(0) = 0$ so that $m(2) = (1 - \exp(-2/\tau))\dot{\theta}^\ast$ where $\dot{\theta}^\ast = [1, -1, 1, -1]^T$ up to time $t = 2$. So DFO recovers the exact crossing continuation. Thus, in this example, this proof sketch shows that DFO recovers the correct PDE trajectory, in the sense that the represented function equals the exact solution for all times t (up to time discretization errors).
>
> - We note however that a universal theorem guaranteeing recovery of the exact PDE solution is beyond the scope of this work and, for nonlinear parametrizations, depends on factors beyond gauge fixing, including manifold expressivity, discretization, truncation, and higher-order effects.
>
> - We will revise the manuscript to provide a detailed proof based on the sketch above and make its scope explicit and note that a general convergence theory for DF(O) remains open.

---

> > ### Author Rebuttal · Reviewer_bzX3 · 2026-04-02
> >
> > I thank the authors for addressing my questions. Their responses are insightful and interesting. I stay with a score of 5.

---

### Official Review · Reviewer_8pKm · 2026-03-11

**Soundness:** 4
**Presentation:** 4
**Significance:** 3
**Originality:** 3
**Overall Recommendation:** 5
**Confidence:** 4

**Summary:**

This paper addresses a core problem of Dirac-Frenkel variational dynamics for neural PDE solvers, where rank deficiency of the parametrization leaves the parameters non-uniquely determined. The authors reframe this non-uniqueness as a gauge freedom and propose fixing it by using an Onsager-type momentum variable exclusively along the nullspace directions, preserving instantaneous residual minimization while promoting temporally smooth parameter trajectories. Numerical experiments across several different PDEs demonstrate that the resulting Dirac-Frenkel-Onsager scheme handles tangent space collapse more robustly than competing methods.

**Compliance With Llm Reviewing Policy:**

Affirmed.

**Key Questions For Authors:**

1) Could it be a problem for the stability of the DFO solution if the null space in equation (8), null(G(theta(t))), varies a lot as a function of t? The projection P(theta(t)) in equation (13) could then also vary a lot. This might be a naive question but in the case of a greatly varying null space, could it be possible that taking the min norm solution of (5) for eta be more stable than using DFO?

2) The projector on the null space depends through the truncated SVD on a parameter epsilon, how sensible are the results to this choice of epsilon ? It seems it could also introduce a bias (similarly to using the min norm solution of (5)) since singular values just above are treated as function-relevant while those below are treated as gauge directions. Any methodology to choose epsilon ?

3) In section 4.4, when choosing the set of collocation in Omega, if I understand correctly this is a standard procedure to choose a grid with a mesh size to discretize Omega. Does the mesh size choice have any incidence on having tagent space collapse or near collapse ? Should the mesh be related to epsilon in some way ?

4) In line 304, col2 you write: Additionally, we consider an ill-conditioned case with ϵ small but positive. Do I understand correctly that this epsilon is tunning how ill-conditioned the wave problem will be, where setting epsilon = 0 would correspond to an exact collapse? In this sense  this epsilon is not the same as the epsilon used for truncating the SVD, but the latter should be chosen in some way based on the former. How does one choose the SVD epsilon in this case ?

5) Typo line 146, col 2: “map” is repeated

**Limitations:**

Yes.

**Strengths And Weaknesses:**

Strength:

This paper has many strengths. First it introduces a physically sound and novel method to perform gauge fixing thus addresing the parametrization problem. It extensively tests the method on numerous cases and demonstrates its performance and practical efficiency. The paper is also well-written and the results well-presented, including many explanations such as the colliding waves illustration. Lastly the paper is also very honest on its limitations which are clearly highlighted.

Weakness:

One weakness seems to be the lack of discussion on how to choose the truncation epsilon which will decide if certain eigenvalues are consider directions in the function space or will be considered too small and included in the null space of the solution. Naively one would think the robustness of the method could depend greatly on this epsilon, and this is not very well explained in the paper. See also the list of questions below.

There are several extra limitations hightlighted by the paper itself, such not including higher-order effects, or not giving a principled method to tune some of the hyperparameters.

Lastly on a slightly different note the paper is more a numerical PDE paper than an ML/AI paper, but this should definitely not be taken as a strong criticism.

---

> ### Author Rebuttal · Authors · 2026-03-30
>
> Thank you for the careful review and for highlighting both the practical value of the method and the importance of being explicit about its limitations.
>
> *"lack of discussion on how to choose the truncation epsilon ... robustness ... could depend greatly on this epsilon [...] how sensible are the results to this choice of epsilon ?"*
> - We agree this should have been discussed more concretely (this comment was also raised by 3egf). It is to a large extent a question about the numerical integration of standard DF itself rather than about our DFO specifically. In practice, the SVD cutoff should be chosen according to what is actually resolvable at the selected time-step / collocation / discretization level: singular directions below that threshold cannot be stably exploited by the time integrator, so treating them as resolved signals is not meaningful.
>
> - We believe a critical question is whether there is a consistent improvement provided by DFO over DF within a reasonable range of SVD truncation levels. To show that this is indeed the case we ran an additional robustness sweep over the SVD truncation levels on the Fokker-Planck equation example (results averaged over five runs):
>
> | SVD truncation level | DF relative error of the mean | DFO relative error of the mean|
> |---|---:|---:|
> | 1e-05 | 7.75e-02 | 6.26e-02 |
> | 5e-05 | 4.73e-02 | 3.99e-03 |
> | 1e-04 | 1.46e-02 | 4.54e-03 |
> | 1e-03 | 1.11e-02 | 6.60e-03 |
> | 1e-02 | 2.46e-02 | 1.88e-02 |
>
> - Over the whole range $[1e-05, 1e-02]$ of truncation values, our DFO approach achieves accuracy improvements over DF. In this sense, these empirical results indicate robustness of DFO.
>
>
>
> *"in the case of a greatly varying null space, could it be possible that taking the min norm solution ... be more stable than using DFO?"*
> - We thank the reviewer for this comment. If the nullspace and its orthogonal complement were fixed in time, then the min-norm DF velocity would always lie in the same complement subspace, and the Onsager history variable $m$ would evolve in that same subspace as well. In that idealized case, its nullspace projection would remain zero, so the DFO correction would vanish and the method would reduce to standard DF. The momentum term becomes active only because the nullspace changes with time. This is exactly the situation we want to stabilize: when the geometry of admissible parameter velocities changes, purely instantaneous gauge selection can become erratic, whereas the filtered history provides a more coherent continuation of the trajectory. The colliding-waves example is already a concrete instance of this phenomenon: the nullspace changes abruptly, from dimension four to dimension two, and this is precisely where incorporating history can be beneficial.
>
> - We will add a discussion about this in the revision of the paper, if accepted.
>
>
> *"Does the mesh size choice have any incidence ... Should the mesh be related to epsilon in some way ?"*
> - The collocation/grid resolution affects how accurately the least-squares problem and its spectrum are represented, so if choosing it too small, one misses information. We confirm this with an experiment on the Fokker-Planck equation, the results of which are reported in the following table. We vary the number of collocation points from 20k to 70k and report the results (averaged over five runs). One can see that if the number of collocation points is chosen too coarse, the accuracy of both DF and DFO goes down (even though DFO still shows an advantage over DF). Going above 30k in this example to 70k points has little effect on the accuracy because 30k points sufficiently resolve this problem for the given time-step size and other discretization parameters.
>
> | num samples | DF relative error of the mean | DFO relative error of the mean |
> |---|---:|---:|
> | 20000 | 2.5e-2 | 1.5e-2 |
> | 30000 | 2.4e-2 | 4.9e-3 |
> | 40000 | 1.5e-2 | 4.5e-3 |
> | 50000 | 9.5e-3 | 4.3e-3 |
> | 60000 | 3.2e-2 | 6.0e-3 |
> | 70000 | 1.5e-2 | 4.2e-3 |
>
>
> *"Do I understand correctly that this epsilon is tuning how ill-conditioned the wave problem will be ... this epsilon is not the same as the epsilon used for truncating the SVD?"*
> - Yes, thank you for pointing this out. This is a clash of notation. We will use a different letter in a revision for the \epsilon referring to the ill-conditioning of the wave problem to avoid confusion.
> - Independent of this, the SVD cutoff epsilon is a numerical threshold used by the solver. The latter should not be chosen from the former by a fixed formula. Instead, it should be chosen based on resolvability and stability at the chosen time step/collocation level. In practice this is also why a nearly singular regime can behave like an effectively singular one.
>
>
> *"Typo ... 'map' is repeated"*
> - Thank you, we will fix this typo.

---

> > ### Author Rebuttal · Reviewer_8pKm · 2026-04-03
> >
> > I thank the authors for having provided a very thorough answer to all of my concerns, including additional numerical experiments to support their answer. I support publication with a score of 5.

---

### Official Review · Reviewer_kk1v · 2026-03-12

**Soundness:** 3
**Presentation:** 2
**Significance:** 3
**Originality:** 3
**Overall Recommendation:** 5
**Confidence:** 3

**Summary:**

The study investigates a failure mode of local-in-time neural PDE solvers based on the Dirac-Frenkel (DF) variational principle, referred to as tangent space collapse. Specifically, when these methods perform instantaneous residual minimization over time, the problem may become ill-conditioned, leading to non-unique parameter velocities. Inappropriate velocity selection criteria may further result in degenerate parameter evolution. The paper interprets this non-uniqueness as a gauge freedom. Based on the Onsager principle, the authors introduce a history variable and inject it along the null-space direction, forming the proposed Dirac-Frenkel-Onsager (DFO) dynamics. This proposed method ensures instantaneous residual minimization while selecting a more stable parameter trajectory. Further, it is evaluated through an toy example, several low-dimensional problems, and a five-dimensional Fokker-Planck equation, demonstrating advantages over the DF-based methods in terms of accuracy and computational cost.

**Compliance With Llm Reviewing Policy:**

Affirmed.

**Final Justification:**

The authors have addressed most of my concerns.

**Key Questions For Authors:**

(1) In the introduction, it would be helpful to clarify the respective advantages and disadvantages of the global-in-time paradigm and the local-in-time paradigm. Providing such a comparison would improve the motivation and context for the proposed method.
(2) The paper repeatedly mentions the issue of non-uniqueness in existing methods. However, in my understanding, as long as a suitable parameter trajectory can be obtained to represent the solution of the dynamical system, non-uniqueness itself may not necessarily constitute a major issue. Therefore, the failure mode studied in this work may require a more in-depth explanation. For example, when parameter trajectories are not unique, inappropriate selection criteria may lead to pathological trajectories (as illustrated in “Example: Colliding waves”). If my understanding is correct, it would be helpful for the authors to provide additional clarification whenever non-uniqueness is discussed throughout the paper, explicitly explaining the issues that arise from it, rather than referring to non-uniqueness in isolation.
(3) This study introduces a history variable to address the considered failure mode. However, could this mechanism conversely introduce new issues? For example, might injecting such a variable cause an otherwise smooth parameter trajectory to fall into one failure mode?

**Limitations:**

Yes

**Strengths And Weaknesses:**

Strengths:
The paper investigates one failure mode in neural PDE solvers and proposes a corresponding remedy, which contributes to a better understanding of neural solvers. Through illustrative examples and analysis, the paper clearly reveals the key issue being addressed. Moreover, the work formalizes the issue in terms of gauge freedom, and the idea of performing gauge fixing within the null space is conceptually novel, demonstrating a notable degree of originality. The experimental evaluation is fairly comprehensive, including one illustrative example that exposes the failure mode, several low-dimensional problems, and a five-dimensional Fokker-Planck equation.
Weaknesses:
Given that the paper introduces a considerable number of concepts and mathematical notations, the presentation could be further improved to enhance readability. For example, in the “Example: Colliding waves” section, it would be helpful to include a pointer to Appendix A or add additional explanatory details to help readers more quickly and clearly understand the key point.

---

> ### Author Rebuttal · Authors · 2026-03-30
>
> Thank you for the thoughtful review and for clearly identifying the main conceptual contribution of the paper.
>
> *“[...] the presentation could be further improved to enhance readability. [...]”*
>
> - We have identified several places where we can improve readability and will do so in a revision. We thank the reviewer for the suggestion on how to improve the readability of the colliding-waves section with a clearer pointer to Appendix A.
>
> *"In the introduction, it would be helpful to clarify the respective advantages and disadvantages of the global-in-time paradigm and the local-in-time paradigm"*
> - This is a good suggestion. Our intent was to position them as complementary rather than competing: global-in-time methods optimize over the whole space-time domain, while local-in-time methods advance the parameters sequentially through local residual minimization. Since our contribution concerns the parameter dynamics induced by Dirac-Frenkel, the local-in-time setting is the natural one here; if accepted, we will make this contrast more explicit in the introduction.
>
> *"non-uniqueness itself may not necessarily constitute a major issue [...] it would be helpful [...] explicitly explaining the issues that arise from it"*
> - Thank you; we agree that we should sharpen the wording in the paper to make this point more clear. We further agree with the reviewer that once multiple parameter velocities produce the same first-order function derivative, a regularizer used to select one representative can lead to pathological parameter trajectories. This is exactly what the colliding-waves example is meant to show: the function-level Dirac-Frenkel condition is satisfied, yet the min-norm selector chooses a parameter trajectory that is qualitatively wrong after collapse. If accepted, we will make this wording sharper and make changes throughout the manuscript.
>
>
> *"This study introduces a history variable to address the considered failure mode. However, could this mechanism conversely introduce new issues?"*
> - We discuss in the paper (see paragraph on “First versus higher-order changes” at the end of page 4) that because nullspace directions are only first-order invisible at the function level, finite step sizes can introduce higher-order drift. We’d like to add though that DFO is first-order correct in the sense that it preserves the instantaneous Dirac--Frenkel (DF) residual-minimizing tangent update. In fact, even standard DF can lead to higher-order drift as it only constrains first-order (tangent-space) residual minimization and imposes no variational control over higher-order effects..

---

> > ### Author Rebuttal · Reviewer_kk1v · 2026-04-03
> >
> > The authors have addressed most of my concerns, so I have increased my score

---

### Official Review · Reviewer_3egf · 2026-03-12

**Soundness:** 4
**Presentation:** 3
**Significance:** 4
**Originality:** 3
**Overall Recommendation:** 5
**Confidence:** 3

**Summary:**

The authors introduce an Onsager gauge principle to resolve jacobian nullspace ambiquities for local-in-time (autoregressive) neural PDEs for time varying problems. They show empirically and analytically that this method does not change the first order residual minimization (unlike simple regularization), and demonstrate strong performance on several canonical PDE problems.

**Compliance With Llm Reviewing Policy:**

Affirmed.

**Final Justification:**

The authors have addressed my concerns, I see this method as useful and novel. I maintain my evaluation (accept).

**Key Questions For Authors:**

Q1. How sensitive are these result to truncation in the SVD?

Q2. Is there guidance on how to select lambda?

**Limitations:**

yes

**Strengths And Weaknesses:**

S1. This paper is well written; the contribution is clear and seemingly novel.

S2. The proposed method is simple to implement, physically motivated, and addresses a known failure mode with dirac-frenkel methods. Most importantly they remedy the issues with standard regularization which modify the minimizer.

S3. The colliding wave example usefully demonstrates this ill-posedness.

S4. The empirical results are strong compared to reasonable baselines.

S5. The limitations are acknowledged and adequately addressed.

Weaknesses:

W1. The history introduces a path-dependence and additional complexity, I would be interested if the authors have considered alternative formulations which avoid this.

W2. As they acknowledge, this method is intended to address exactly singular nullspaces, however it is not clear how common these issues are in practical examples compared to nearly singular conditioning. The empirical results do support the generality of this method however.

---

> ### Author Rebuttal · Authors · 2026-03-30
>
> Thank you for the careful reading and for highlighting both the conceptual novelty and the empirical strength of the paper.
>
> *"The history introduces a path-dependence and additional complexity, I would be interested if the authors have considered alternative formulations which avoid this."*
> - ​​We agree that introducing the history variable adds one auxiliary state and therefore some path dependence. More generally, Eq. (9) was meant precisely as a flexible gauge-fixing template, so other formulations (including potentially history-free choices) are possible within this template and are an interesting direction for future work. We focus in this paper on the Onsager choice because it yields (i) a simple closed-form update and (ii) adds very little computational overhead (as discussed in Section 4.5 and as reflected in the runtime results in, e.g., Table 1-2), and gives the auxiliary variable a (iii) clear interpretation as a momentum-like exponential moving average of past reference velocities, while still modifying the dynamics only through nullspace directions and hence (iv) preserving the instantaneous Dirac-Frenkel residual-minimizing update.
>
> - We will revise the manuscript to emphasize that the template given in (9) can be applied with other gauge fixing techniques.
>
> *"this method is intended to address exactly singular nullspaces ... not clear how common these issues are in practical examples compared to nearly singular conditioning. The empirical results do support the generality of this method however."*
> - Our view is that, in finite precision, sufficiently ill-conditioned directions behave practically as a nullspace. Once singular values fall below the resolvable level set by discretization/collocation/time integration, the numerical behavior is effectively the same failure mode as exact singular nullspaces. This is why we discuss both exact nullspaces and numerical nullspaces/tangent-space collapse in ill-conditioned regimes. As the reviewer points out, we do see an improved accuracy also in these ill-conditioned regimes in our numerical experiments.
>
> *"How sensitive are these results to truncation in the SVD?"*
> - This is an important question, but it is to a large extent a question about the numerical integration of standard DF itself rather than about our DFO specifically. In practice, the SVD cutoff should be chosen according to what is actually resolvable at the selected time-step / collocation / discretization level: singular directions below that threshold cannot be stably exploited by the time integrator, so treating them as resolved signals is not meaningful.
>
> - We believe a critical question is whether there is a consistent improvement provided by DFO over a reasonable range of SVD truncation levels. To show that this is indeed the case we ran an additional robustness sweep over the SVD truncation levels on the Fokker-Planck equation example (results averaged over five runs):
>
> | SVD truncation level | DF relative error of the mean | DFO relative error of the mean|
> |---|---:|---:|
> | 1e-05 | 7.75e-02 | 6.26e-02 |
> | 5e-05 | 4.73e-02 | 3.99e-03 |
> | 1e-04 | 1.46e-02 | 4.54e-03 |
> | 1e-03 | 1.11e-02 | 6.60e-03 |
> | 1e-02 | 2.46e-02 | 1.88e-02 |
>
> - Over the whole range $[1e-05, 1e-02]$ of truncation values, our DFO approach achieves accuracy improvements over DF. In this sense, these empirical results indicate robustness of DFO.
>
>
> *"Is there guidance on how to select $\lambda$?"*
> - Because $\lambda$ is the weight with which the gauge momentum is added we expect it to be on the order of 1, otherwise the gauge part is much over-/under-weighted compared to the residual term. In fact, in all our examples, $\lambda$ is in the range $[0.5,2.5]$. A specific \lambda can be found with a grid search with, e.g., the residual norm as objective. Furthermore, the following table shows a lambda-robustness analysis for the Fokker-Plank equation example over $[0.5, 2.5]$. It can be seen that the results are fairly stable over this reasonable $\lambda$ range (results averaged over five runs).
>
> | $\lambda$ | DFO relative error of the mean|
> |---:|---:|
> | 5.00e-1 | 1.36e-2 |
> | 7.50e-1 | 1.53e-2 |
> | 1.00e0 | 2.20e-2 |
> | 1.25e0 | 1.65e-2 |
> | 1.50e0 | 7.76e-3 |
> | 1.75e0 | 1.40e-2 |
> | 2.00e0 | 8.53e-3 |
> | 2.25e0 | 3.51e-2 |
> | 2.50e0 | 7.75e-3 |

---

> > ### Author Rebuttal · Reviewer_3egf · 2026-04-02
> >
> > I appreciate the thorough rebuttal which has helped clarify some aspects of this work.

---

### Decision · Program_Chairs · 2026-04-30

**Decision:**

Accept (spotlight)

**Comment:**

This paper makes a strong contribution to the study of local-in-time neural PDE solvers by identifying a meaningful failure mode in Dirac–Frenkel dynamics and proposing a principled remedy based on gauge fixing through an Onsager-type momentum term. Reviewers were uniformly positive about the paper’s novelty, technical quality, and empirical support, highlighting in particular the clear formulation of the non-uniqueness issue, the appealing interpretation in terms of gauge freedom, and the strong experimental validation across illustrative and higher-dimensional PDE examples. The main concerns raised in the reviews were about presentation, the role and tuning of the SVD truncation threshold and other hyperparameters, and the scope of the theoretical justification. In the author–reviewer discussion, the authors addressed these concerns well: they clarified why non-uniqueness is problematic when it leads to pathological parameter trajectories, provided additional robustness analyses for the truncation threshold, lambda, and collocation resolution, and gave a more concrete explanation of the theoretical scope, including a proof sketch for exact recovery in the colliding-waves example. All reviewers indicated that their concerns were fully resolved after rebuttal, with one reviewer explicitly increasing their score and another explicitly stating support for publication. Overall, I find the paper technically solid, original, and well executed, and I believe it is important to include in the program given both the quality of the work and the significance of the contribution.